# TRAINING-FREE REASONING AND REFLECTION IN MLLMS

## ABSTRACT

Recent advances in Reasoning LLMs (e.g., DeepSeek-R1 and OpenAI-o1) have showcased impressive reasoning capabilities via reinforcement learning. However, extending these capabilities to Multimodal LLMs (MLLMs) is hampered by the prohibitive costs of retraining and the scarcity of high-quality, verifiable multimodal reasoning datasets. This paper introduces **ANO** Model (a temporal name for anonymous review), a training-free and R1-like MLLM that imbues off-the-shelf MLLMs with reasoning and reflection abilities, without any gradient updates or extra supervision. Our key insight is to decouple perception and reasoning across MLLM decoder layers. Specifically, we observe that compared to the deeper decoder layers, the shallow decoder layers allocate more attention to visual tokens, while the deeper decoder layers concentrate on textual semantics. This observation motivates a hierarchical weight merging approach that combines a visual-pretrained MLLM with a reasoning-specialized LLM. To this end, we propose a layer-wise, Taylor-derived closed-form fusion mechanism that integrates reasoning capacity into deep decoder layers while preserving visual grounding in shallow decoder layers. Extensive experiments on challenging multimodal reasoning benchmarks demonstrate the effectiveness of our approach. On the MMMU benchmark, our model ANO-38B achieves an accuracy of 69.2, outperforming the strongest baseline InternVL2.5-38B by +5.3, and even surpasses the proprietary GPT-4o model.

## 1 INTRODUCTION

Recent reasoning-focused large language models (LLMs) (Guo et al., 2025; Team et al., 2025b; Ma et al., 2025; Lightman et al., 2024) such as DeepSeek-R1 (Guo et al., 2025) and OpenAI-o1 (OpenAI, 2024b) have demonstrated strong performance in tasks requiring complex logic, including math reasoning, symbolic manipulation, and program synthesis. These models leverage mechanisms like reinforcement learning to perform multi-step problem-solving and iterative self-correction, often surpassing even human experts.

In real-world scenarios, numerous tasks demand sophisticated multimodal reasoning capabilities. For instance, solving visual mathematics problems, interpreting diagrams, and understanding code snippets embedded within images require the integration of visual perception with logical reasoning. Inspired by the successes of reasoning-augmented LLMs, researchers have begun exploring methods to endow multimodal large language models (MLLMs) (Xu et al., 2024; Dong et al., 2025; Yao et al., 2024; Liu et al., 2025) with similar reasoning abilities. A prevalent approach (Huang et al., 2025; Peng et al., 2025; Zhang et al., 2025) involves adapting reinforcement learning techniques, such as GRPO (Guo et al., 2025), to the multimodal context. However, this strategy encounters significant challenges. Firstly, the reinforcement learning training of large-scale MLLMs demands substantial computational resources, making it resource-intensive. Secondly, there is a notable scarcity of high-quality, verifiable multimodal reasoning datasets, which are essential for effective training. This paucity of suitable data severely impedes the development and scalability of reasoning-capable MLLMs.

In this paper, we introduce **ANO**, a training-free and R1-like MLLM that is designed to endow existing MLLMs with advanced reasoning and reflection capabilities without any additional training or supervision. Figure 1 visualizes this pipeline. Our method is built upon two key insights:

Figure 1: Non-reasoning MLLMs lack reasoning and reflection abilities, while reasoning LLMs are unable to perceive visual information. We propose a training-free, closed-form layerwise fusion method that combines visual perception and language reasoning strengths, substantially enhancing overall reasoning capability in multimodal settings.

- **Cross-modal Model Merging**: We conceptualize MLLMs as base LLMs fine-tuned on visual-text data, while reasoning-specialized LLMs represent the same base LLM fine-tuned on reasoning tasks. According to the task arithmetic hypothesis (Wortsman et al., 2022; Ilharco et al., 2023; Zhang et al., 2023), the difference in weights between a fine-tuned model and its base model encapsulates the task-specific adaptations. By merging task vectors from models fine-tuned on different tasks, we can integrate multiple capabilities into a single model without additional training.
- **Layer-wise Functional Specialization in MLLMs**: Drawing inspiration from the hierarchical processing observed in the human brain (Brincat et al., 2018; Kawasaki et al., 2022), where sensory inputs are initially processed in primary sensory areas and progressively integrated into higher-order cognitive functions in association cortices, we observe a similar pattern in MLLMs. As shown in Figure 2 (a), compared to the deeper decoder layers, the shallow decoder layers allocate more attention to visual tokens, facilitating perceptual grounding, while the deeper decoder layers concentrate on textual semantics.

Based on these two key insights, we design a hierarchical weight merging strategy to effectively integrate a vision-text pretrained MLLM with a reasoning-specialized LLM. Specifically, we develop a layer-wise, Taylor-derived closed-form fusion mechanism that enables fine-grained control over the contribution of each model at different depths of the decoder. This mechanism builds on the task vector formulation and refines it with a layer-wise optimization strategy: For each decoder block, we derive a closed-form solution for the optimal task vector fusion weights by minimizing the Taylor-approximated task loss difference. This allows precise control over how visual and reasoning adaptations are combined at each layer. Furthermore, guided by the empirical prior that shallow decoder layers in MLLMs attend more to visual inputs while deeper layers focus on symbolic reasoning, we incorporate layer-dependent fusion weights to align with the distinct functional roles across the model hierarchy. This design enables ANO to embed reasoning capabilities into deeper layers, responsible for abstraction and reasoning, while preserving the visual grounding in shallower layers, which are more sensitive to perceptual signals.

We validate the effectiveness of ANO through comprehensive evaluations on challenging multimodal reasoning benchmarks. Notably, our model ANO-38B achieves an accuracy of 69.2 on the MMMU benchmark, outperforming the strongest baseline InternVL2.5-38B by +5.3, and even surpasses the proprietary GPT-4o model. These results underscore ANO's ability to enhance reasoning capabilities in MLLMs without additional training or supervision.

In summary, our contributions are threefold:

- The layer-wise functional specialization in MLLMs, where shallow decoder layers focus on visual perception and deeper layers on textual reasoning, is identified and leveraged.

- A novel, training-free hierarchical weight fusion mechanism is proposed, integrating reasoning capabilities into existing MLLMs by merging task vectors at each layer, guided by a Taylor-derived closed-form solution.

- It is demonstrated that ANO effectively enhances the reasoning and reflection abilities of MLLMs, achieving superior performance on multimodal reasoning tasks without the need for additional training or supervision.

## 2 RELATED WORK

**Multimodal Large Language Models.** Multimodal large language models (MLLMs) integrate visual and linguistic understanding via large-scale vision-language pretraining and instruction tuning (Wang et al., 2023; Bai et al., 2023; Liu et al., 2023; Sun et al., 2024), enabling zero- or few-shot VQA, image captioning, and multimodal instruction following. While effective at aligning visual and textual features, their ability for multi-step reasoning or self-reflection remains limited. Inspired by chain-of-thought prompting in text LLMs (Wei et al., 2022), recent works (Xu et al., 2024; Dong et al., 2025; Yao et al., 2024; Liu et al., 2025; Huang et al., 2025; Peng et al., 2025; Zhang et al., 2025) aim to enhance reasoning in MLLMs. Simple CoT prompts often bring marginal gains (Zhang et al., 2024), whereas plan-based approaches like CCoT (Mitra et al., 2024) and TextCoT (Luan et al., 2024) guide structured reasoning. Reinforcement-learning extensions such as MM-Eureka (Meng et al., 2025), Vision-R1 (Huang et al., 2025), and LMM-R1 (Peng et al., 2025) further improve reasoning but require expensive fine-tuning on scarce multimodal reasoning datasets, limiting scalability. Moreover, recent studies (Chen et al., 2024a) observe uneven allocation of visual attention across shallow and deep layers in MLLMs, which has been exploited mainly for visual token pruning. In contrast, we leverage this phenomenon to guide a layer-wise fusion strategy between vision and reasoning models, enabling training-free integration of perception and long-CoT reasoning.

**Model Merging.** Model merging (Matena & Raffel, 2022; Wortsman et al., 2022; Ilharco et al., 2023; Yadav et al., 2023; Zhou et al., 2024; Jin et al., 2023) efficiently fuses multiple task-specific models without accessing original training data. Early weight averaging (Wortsman et al., 2022) improved multi-task robustness, while Task Arithmetic (Ilharco et al., 2023) introduced task vectors to combine fine-tuned weights. Methods like TIES-Merging (Yadav et al., 2023) and DARE (Yu et al., 2024) reduce task interference via pruning, alignment, and rescaling of updates. MetaGPT (Zhou et al., 2024) frames merging as multi-task learning for balanced performance. These approaches work well for text-only LLMs but do not fully leverage multimodal model structures.

## 3 APPROACH

In this section, we introduce ANO, a training-free and R1-like MLLM that endows off-the-shelf MLLMs with advanced reasoning and reflection by merging them with reasoning-specialized LLMs, entirely without gradient updates or extra supervision.

### 3.1 PRELIMINARY: TASK ARITHMETIC FOR MODEL MERGING

We build on the paradigm of task arithmetic (Ilharco et al., 2023), which provides a simple yet effective mechanism to merge multiple fine-tuned models, so-called homologous models, that share the same base architecture. Let $f(\cdot\,; \theta_0)$ denote a pre-trained base model, and let

$$\theta_t = \arg\min_\theta \mathcal{L}_t\big(f(\cdot\,; \theta), \mathcal{D}_t\big) \tag{1}$$

be the parameters obtained by fine-tuning on task $t$ with loss $\mathcal{L}_t$ over dataset $\mathcal{D}_t$. We define the task vector for task $t$ as $\tau_t = \theta_t - \theta_0$. Under the homologous assumption that all $\theta_t$ lie in the same parameter space as $\theta_0$, we can form a merged model by a linear combination of these task vectors:

$$\theta_f = \theta_0 + \sum_{t \in \{V, R\}} \lambda_t\, \tau_t\,, \tag{2}$$

where $\{\lambda_t\}$ are non-negative fusion weights controlling each task's contribution.

Early approaches choose all $\lambda_t$ heuristically (e.g., a constant 0.3) or via grid search on held-out data, but these methods either under-utilize model capacity or incur prohibitive search costs as $T$ grows.

### 3.2 ANO: A TRAINING-FREE AND R1-LIKE MLLM

Building on the task-arithmetic preliminary, ANO introduces two key innovations: **Layer-wise Fusion**, which respects the functional specialization of each decoder layer (for clarity, we define a decoder's block as a "layer"); **Modality Priors**, which steer shallow layers toward visual grounding and deep layers toward symbolic reasoning.

### 3.2.1 Layer-Wise Fusion Setup for MLLMs

ANO bridges these two paradigms by viewing the vision-fine-tuned decoder and a reasoning-specialized decoder as variants of the same base model checkpoint. Leveraging their shared architecture, we can merge their strengths without retraining. The decoder consists of $L$ stacked transformer layers, indexed by $l = 1, \ldots, L$. Figure 2 (a) shows that, compared to the deeper decoder layers, the shallow decoder layers allocate more attention to visual tokens, facilitating perceptual grounding, while the deeper decoder layers concentrate on textual semantics. To preserve this functional hierarchy and prevent interference between vision and reasoning, ANO performs layer-wise weight fusion: each decoder layer is merged independently, preserving its specialized role.

Concretely, let $\theta_0^{(l)}, \theta_V^{(l)}, \theta_R^{(l)}$ denote the parameters of layer $l$ for the pre-trained base model, the vision-fine-tuned MLLM, and the reasoning-fine-tuned LLM, respectively. We define the layer-wise task vectors:

$$\tau_V^{(l)} = \theta_V^{(l)} - \theta_0^{(l)}, \qquad \tau_R^{(l)} = \theta_R^{(l)} - \theta_0^{(l)}. \tag{3}$$

These differences capture how each task (vision vs. reasoning) shifts the model weights at each depth.

To measure the impact of fusing these shifts, we introduce two metrics. First, the Layer-Wise Task Loss Difference (LTLD) compares the fused layer's loss against each branch's own fine-tuned layer:

**Definition 1** (Layer-Wise Task Loss Difference, LTLD). *Let $\mathcal{L}_t^{(l)}(\theta, x)$ be the loss of branch $t \in \{V, R\}$ at layer $l$ on input $x$. For fusion weights $(\lambda_V, \lambda_R)$, define the fused parameter*

$$\theta_f^{(l)} = \theta_0^{(l)} + \lambda_V \, \tau_V^{(l)} + \lambda_R \, \tau_R^{(l)}, \tag{4}$$

*where, $\theta_f^{(l)}$ represents the parameters of the fusion model at layer $l$, while $\tau_V^{(l)}$ and $\tau_R^{(l)}$ denote the task vectors at layer $l$ of the non-reasoning MLLM and reasoning-specialized LLM, respectively. Then, we define the layer-wise task loss difference (LTLD) as*

$$\mathrm{LTLD}_t^{(l)}(\lambda_V, \lambda_R) = \mathcal{L}_t^{(l)}(\theta_f^{(l)}, x) - \mathcal{L}_t^{(l)}(\theta_b^{(l)}, x), \tag{5}$$

*quantifies the degradation when using the fused weights in place of branch $b$'s own.*

**Definition 2** (Layer-Wise Average Loss Difference, LALD). *The LALD averages LTLD across both branches,*

$$\mathrm{LALD}^{(l)}(\lambda_V, \lambda_R) = \tfrac{1}{2}\big(\mathrm{LTLD}_V^{(l)} + \mathrm{LTLD}_R^{(l)}\big), \tag{6}$$

*and serves as our per-layer fusion objective. By minimizing LALD independently at each layer, we derive fusion weights that optimally balance visual grounding and reasoning without cross-layer interference.*

We emphasize that LTLD is a theoretical sensitivity measure, not an operation we perform in implementation. Specifically, LTLD represents the second-order Taylor approximation of the loss increase induced by small perturbations in the parameters of layer $l$, holding all other layers fixed. Under the Neural Tangent Kernel (NTK) (Jacot et al., 2018) linearization and task-vector orthogonality assumptions, this localized view yields a tractable quadratic form in the norms $|\tau_V^{(l)}|^2$ and $|\tau_R^{(l)}|^2$, from which we derive closed-form fusion weights (see Section 3.2.2). Importantly, we do not compute per-layer losses on the fly; LTLD merely guides our analytical derivation. In subsequent sections, we will show how to approximate LTLD via a second-order Taylor expansion, invoke NTK linearization and vector orthogonality, and obtain a data-agnostic closed-form for $(\lambda_V^{(l)}, \lambda_R^{(l)})$.

### 3.2.2 Taylor-Based Approximation and Closed-Form Fusion Weights

To obtain efficient, data-agnostic fusion weights at each decoder layer, we approximate the layer-wise average loss difference (LALD) using a second-order Taylor expansion. Intuitively, this expansion captures how a small perturbation—arising from merging vision and reasoning task vectors—affects the loss. Under two standard assumptions:

**NTK Linearization.** In the infinite-width regime, neural networks evolve under training according to a fixed NTK (Jacot et al., 2018), which implies that small weight perturbations produce locally linear changes in the model output (Jacot et al., 2018; Zhou et al., 2024). The appendix A.15 provides additional details. Previous study (Zhou et al., 2024) has empirically validated this NTK-linear

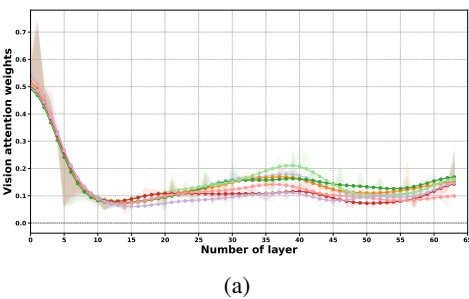
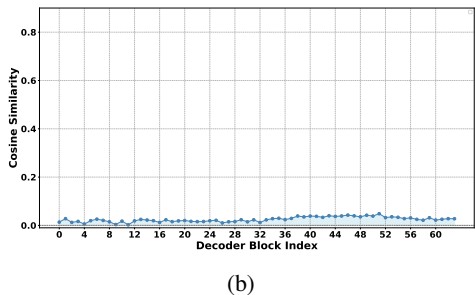

(a)                                                                 (b)

Figure 2: **(a)** Layer-wise visual attention of InternVL2.5-38B. Each curve shows the average attention from a text token to all visual tokens across layers. Shallow layers assign significantly higher attention to visual tokens, while attention in deeper layers approaches zero and rapidly descends (with text attention defined as $1-$ visual attention), indicating a shift from perception to language reasoning. This supports our use of an exponential decay prior to the fusion process. **(b)** Cosine similarity between task vectors of vision-finetuned (InternVL2.5-38B) and reasoning-finetuned (QwQ-32B) models at each decoder block. The task vector at each block is computed by flattening the weight deltas with respect to the base model. The similarity remains close to 0 across all layers, indicating strong near-orthogonality.

behavior for LLMs. They evaluated LLaMA-2-7b-chat-hf (Touvron et al., 2023) on the AGIEval benchmark (Zhong et al., 2024), sampling three random prompts and measuring model outputs for interpolation coefficients $\alpha \in \{0, 0.1, \ldots, 1.0\}$. The output trajectories scale almost perfectly linearly with $\alpha$, confirming that LLM operates in the NTK regime during fine-tuning, which is specifically suitable for the LLM's arithmetic scenario.

**Task-Vector Orthogonality.** Although both vision and reasoning fine-tuning update the same decoder weights, their resulting task vectors often lie in nearly orthogonal subspaces. As shown in Figure 2 (b), we verify this by computing the cosine similarity between $\tau_V^{(l)}$ and $\tau_R^{(l)}$ at each layer, which remains close to zero across all layers. The appendix A.16 provides additional details.

Under the NTK linearization and task-vector orthogonality assumptions, we can now quantify how merging vision and reasoning updates perturbs the layer loss. In the next step, we expand each task's loss around its fine-tuned parameters to derive a closed-form bound on the loss increase.

Let $\theta_0^{(l)}$ be the initialization at layer $l$, and $\theta_t^{(l)} = \theta_0^{(l)} + \tau_t^{(l)}$ be the fine-tuned weights for task $t \in \{V, R\}$. Define the fused weights

$$\theta_f^{(l)} = \theta_0^{(l)} + \sum_{t \in \{V,R\}} \lambda_t^{(l)} \tau_t^{(l)} \tag{7}$$

and the fusion residual

$$h_t^{(l)} = \theta_f^{(l)} - \theta_t^{(l)} = \sum_{k \neq t \in \{V,R\}} \lambda_k^{(l)} \tau_k^{(l)} - (1 - \lambda_t^{(l)}) \tau_t^{(l)}. \tag{8}$$

Around $\theta_t^{(l)}$, a second-order Taylor expansion of the layer loss $\mathcal{L}_t^{(l)}$ gives

$$\mathcal{L}_t^{(l)}(\theta_f^{(l)}) \approx \mathcal{L}_t^{(l)}(\theta_t^{(l)}) + \underbrace{\nabla \mathcal{L}_t^{(l)}(\theta_t^{(l)})^\top h_t^{(l)}}_{\approx 0} + \tfrac{1}{2} h_t^{(l)\top} \nabla^2 \mathcal{L}_t^{(l)}(\theta_t^{(l)}) h_t^{(l)}, \tag{9}$$

where the first-order term vanishes under near-convergence, as guaranteed by the NTK theory for wide networks. This NTK regime further implies two key properties for the second-order term: 1) The Hessian is dominated by the Jacobian Gram matrix due to the approximate linearity of the network in parameter space, and 2) Its eigenvalue distribution becomes approximately isotropic. Concretely, the Hessian admits the following approximation:

$$\nabla^2 \mathcal{L}_t^{(l)}(\theta_t^{(l)}) \approx \delta_t^{(l)} I_{d_l}, \quad \delta_t^{(l)} = \tfrac{1}{d_l} \mathrm{tr}\Big( \nabla f^{(l)}(x_t; \theta_0^{(l)}) \nabla f^{(l)}(x_t; \theta_0^{(l)})^\top \Big), \tag{10}$$

where, $\delta_t^{(l)}$ is a data-dependent constant, $d_l$ represents the parameter dimension of the $l$-th layer, $I_{d_l}$ denotes the $d_l$-dimensional identity matrix, which is used to simplify the structure of the Hessian. Consequently,

$$\text{LTLD}_t^{(l)} = \mathcal{L}_t^{(l)}(\theta_f^{(l)}) - \mathcal{L}_t^{(l)}(\theta_t^{(l)}) \leq \frac{\delta_t^{(l)}}{2} \|h_t^{(l)}\|^2. \tag{11}$$

Summing the above bound for $t \in \{V, R\}$ and using $\langle \tau_V^{(l)}, \tau_R^{(l)} \rangle \approx 0$ yields a layer-wise bound on the average loss increase:

$$\text{LALD}^{(l)} \leq \frac{1}{2} \sum_{t \in \{V,R\}} \delta_t^{(l)} \left[ (1 - \lambda_t^{(l)})^2 \|\tau_t^{(l)}\|^2 + \sum_{k \neq t \in \{V,R\}} (\lambda_k^{(l)})^2 \|\tau_k^{(l)}\|^2 \right] \|\tau_t^{(l)}\|^2. \tag{12}$$

To derive the optimal fusion weights, we minimize the LALD bound under the constraint $\lambda_V + \lambda_R = 1$. Setting the derivative to zero and solving yields the closed-form solution:

$$\lambda_t^{(l)} = \frac{\|\tau_t^{(l)}\|^2}{\|\tau_V^{(l)}\|^2 + \|\tau_R^{(l)}\|^2}, \quad t \in \{V, R\}. \tag{13}$$

This compact expression emphasizes ANO's key advantage: fusion weights depend solely on the observed parameter shifts of each task, eliminating the need for held-out data, grid search, or additional training. For full statements of the Taylor expansion lemma, formal properties, and proofs leading to this closed form, please refer to Appendix A.17 and A.18.

### 3.2.3 INCORPORATING MODALITY PRIORS

While the closed-form fusion weights from Section 3.2.2 balance visual and reasoning shifts purely by their magnitudes, MLLM decoders exhibit a clear functional hierarchy. To encode this prior knowledge, we introduce layer-dependent modality priors $w_V^{(l)}, w_R^{(l)} > 0$ and reformulate the per-layer fusion objective:

$$\min_{\lambda_V, \lambda_R} w_V^{(l)} \text{LTLD}_V^{(l)}(\lambda_V, \lambda_R) + w_R^{(l)} \text{LTLD}_R^{(l)}(\lambda_V, \lambda_R). \tag{14}$$

The modality prior-guided closed-form is

$$\lambda_t^{(l)} = \frac{w_t^{(l)} \|\tau_b^{(l)}\|^2}{w_V^{(l)} \|\tau_V^{(l)}\|^2 + w_R^{(l)} \|\tau_R^{(l)}\|^2}, \quad t \in \{V, R\}. \tag{15}$$

**Attention-Guided Decay Priors** During our experiments, we observed that compared to deeper decoder layers, shallower decoder layers in the MLLM allocate more attention to visual tokens, thereby facilitating visual perceptual grounding, while deeper layers focus primarily on textual semantics, as illustrated in Figure 2 (a). To introduce reasoning and reflection capabilities while preserving visual perception, and to maintain methodological simplicity without introducing additional supervision, we derive modality priors from the model's layer-wise attention patterns and fit them with an exponential decay function (prioritizing simplicity, though more complex quadratic functions could alternatively be used for fitting). This approach naturally captures the non-uniform transition from visual grounding to symbolic reasoning observed in practice. Specifically, we first collect the visual attention weights $a_l$ from each decoder layer in the MLLM. Then, we posit $a_l \approx C\,e^{-\hat{\alpha}\,l}$, and obtain $\hat{\alpha}, \log C$ via linear regression on $\{(l, \log a_l)\}_{l=1}^{L}$. Finally, we set

$$w_V^{(l)} = \frac{\exp(-\hat{\alpha}\,l)}{\sum_{j=1}^{L} \exp(-\hat{\alpha}\,j)}, \qquad w_R^{(l)} = 1 - w_V^{(l)}. \tag{16}$$

This attention-guided exponential schedule requires no labels and ensures the modality priors faithfully mirror the model's intrinsic shift from visual grounding to reasoning across the decoder hierarchy. Please refer to Appendix A.19 for details.

To conclude, ANO delivers a training-free, interpretable, and efficient framework for the per-layer fusion of visual grounding and logical reasoning in MLLMs, requiring only task-vector norms and simple priors. Its closed-form weights avoid any extra labeling data or optimization, making ANO practical for scaling multimodal intelligence.

Table 1: Comparison of ANO variants (8B, 15B, 38B) and state-of-the-art baselines across four multimodal reasoning benchmarks: MMMU val, MMMU-Pro standard (10 opts), MathVision testmini, and WeMath testmini. * indicates the baseline model.

| | Methods | Size | MMMU | MMMU-Pro | MathVision | WeMath |
|---|---|---|---|---|---|---|
| Samll | LLaVA-1.5 (Liu et al., 2024a) | 7B | 35.7 | 19.7 | 10.2 | 7.0 |
| | LLaVA-NeXT (Liu et al., 2024b) | 7B | 35.3 | 19.4 | 10.0 | 3.3 |
| | LLaVA-LLaMA3 (Liu et al., 2024b) | 8B | 39.2 | - | - | - |
| | VILA1.5-LLaMA3 (Lin et al., 2024) | 8B | 38.6 | - | - | - |
| | Idefics3* (Laurençon et al., 2024) | 8B | 43.9 | 32.6 | 20.1 | **12.3** |
| | **ANO** (Ours) | 8B | **48.3** | **34.7** | **27.6** | 11.7 |
| Medium | LLaVA-1.5 (Liu et al., 2024a) | 13B | 37.0 | - | 13.1 | 7.4 |
| | ShareGPT4V (Chen et al., 2024b) | 13B | 36.6 | - | 13.9 | - |
| | LLaVA-NeXT (Liu et al., 2024b) | 13B | 36.2 | 19.8 | - | - |
| | VILA-1.5 (Lin et al., 2024) | 13B | 37.9 | - | 15.2 | 11.4 |
| | NVILA* (Liu et al., 2024c) | 15B | 53.2 | 36.2 | 23.2 | 31.1 |
| | **ANO** (Ours) | 15B | **61.3** | **49.4** | **37.2** | **32.3** |
| Large | LLaVA-NeXT (Liu et al., 2024b) | 34B | 48.1 | 30.3 | - | - |
| | VILA-1.5 (Lin et al., 2024) | 40B | 51.9 | 35.9 | - | - |
| | LLaVA-OV (Li et al., 2024) | 72B | 56.8 | 38.0 | 25.3 | 32.0 |
| | Qwen2-VL (Wang et al., 2024b) | 72B | 64.5 | 49.2 | 26.6 | 36.0 |
| | GPT-4o (OpenAI, 2024a) | - | 69.1 | 54.0 | 29.9 | - |
| | InternVL2.5* (Chen et al., 2024c) | 38B | 63.9 | 48.0 | 32.2 | 38.3 |
| | **ANO** (Ours) | 38B | **69.2** | **56.8** | **39.7** | **47.0** |

## 4 EXPERIMENTS

### 4.1 DATASETS AND MODEL VARIANTS

We evaluate ANO on four widely-used multimodal reasoning benchmarks: MMMU val (Yue et al., 2024a), MMMU-Pro standard (10 opts) (Yue et al., 2024b), MathVision testmini (Wang et al., 2024a), and WeMath testmini (Qiao et al., 2024). These datasets encompass a broad spectrum of task formats and difficulty levels.

To evaluate the effects of scale and architecture, we construct three ANO variants via layer-wise fusion of non-reasoning MLLMs and reasoning LLMs: **ANO-8B** (Idefics3-8B + DeepSeekDistil-LLaMA3-8B), **ANO-15B** (NVILA-15B + DeepSeekDistil-Qwen2.5-14B), and **ANO-38B** (InternVL2.5-38B + QwQ-32B). These cover LLaMA-based and Qwen-based decoders with different capacities. Notably, the modality prior is robust across distinct architectures: **ANO-8B** employs a SigLIP encoder with a LLaMA3.1 decoder, **ANO-15B** employs a SigLIP encoder with a Qwen2.5 decoder, and **ANO-38B** employs an InternViT-6B backbone with a Qwen2.5 decoder. Despite these differences, all variants consistently exhibit exponential decay of visual attention across decoder layers, confirming the universality of our prior beyond specific model families.

We apply ANO's closed-form fusion weights layer-by-layer without any additional fine-tuning. The attention-guided exponential decay prior is fit on 1000 randomly sampled validation examples from general-domain dataset (**note: only the image is needed, no annotations**) using least-squares regression to obtain decay parameter $\alpha$ and normalization constant $C$. We adopt MSCOCO (Lin et al., 2014) as a representative general-domain image dataset, since it is widely accessible and commonly used for evaluating vision models. Importantly, our approach only requires generic images (without annotations), and thus any large-scale, domain-agnostic image collection could serve this purpose. Moreover, in Appendix A.7, we compared attention decay curves collected from both general-domain validation data and domain-specific validation sets, and found that the latter only brings marginal improvements within the target domain while negatively impacting performance on other domains.

### 4.2 QUANTITATIVE EVALUATION

**MMMU/MMMU-Pro.** On MMMU (college-level image-text questions), ANO-8B achieves 48.3, a 4.4-point gain over its vision branch. This improvement indicates that even at 8B parameters, our

Table 2: Ablation study results of ANO-15B across four multimodal reasoning benchmarks: MMMU val, MMMU-Pro standard (10 opts), MathVision testmini, and WeMath testmini. * indicates the baseline model.

| Methods | MMMU | MMMU-Pro | MathVision | WeMath | Average |
|---|---|---|---|---|---|
| NVILA* (Baseline) | 53.2 | 36.2 | 23.2 | 31.1 | 35.9 |
| VLM-Merging (Chen et al., 2025) [ICML'25] | 53.6 | 36.5 | 24.5 | 31.1 | 36.4 |
| AdaMMS (Du et al., 2025) [CVPR'25] | 53.8 | 36.9 | 25.1 | 31.4 | 36.8 |
| Task Arithmetic (Ilharco et al., 2023) [ICLR'23] | 56.1 | 40.1 | 26.7 | 31.3 | 38.6 |
| MetaGPT (Zhou et al., 2024) [EMNLP'24] | 57.9 | 41.2 | 27.9 | 31.3 | 39.6 |
| ANO-15B w/o MP (Ours) | 58.4 | 41.8 | 28.7 | 31.4 | 40.1 |
| **ANO-15B (Ours)** | **61.3** | **49.4** | **37.2** | **32.3** | **45.1** |

Table 3: Perceptual performance of ANO-15B on MME benchmark. "w/o MP" denotes the ablation without Modality Prior. Sub-task abbreviations: "Comm." = Commonsense Reasoning, "Num." = Numerical Calculation, "Text." = Text Translation, "Code." = Code Reasoning.

| MME | Existence | Count | Position | Color | OCR | Poseter | Celebrity |
|---|---|---|---|---|---|---|---|
| MetaGPT | 95.0 | 76.7 | 65.0 | 88.3 | 80.0 | 90.0 | 75.7 |
| ANO-15B w/o MP | 96.7 | 78.3 | **68.3** | 90.0 | 75.0 | 91.2 | 76.2 |
| ANO-15B | **96.7** | **80.0** | 60.0 | **91.7** | **85.0** | **91.8** | **77.7** |
| Perceptual upper bound | 100.0 | 80.0 | 86.7 | 91.7 | 90.0 | 94.2 | 83.2 |

| MME | Scene | Landmark | Artwork | Comm. | Num. | Text. | Code. |
|---|---|---|---|---|---|---|---|
| MetaGPT | 79.8 | 81.7 | **79.0** | 84.0 | 70.0 | 47.5 | 85.0 |
| ANO-15B w/o MP | 80.0 | 83.7 | 76.5 | 84.3 | **75.0** | 57.5 | 82.5 |
| ANO-15B | **83.0** | **84.3** | 74.3 | **85.0** | 72.5 | **60.0** | **85.0** |
| Perceptual upper bound | 83.5 | 90.5 | 82.3 | 82.9 | 72.5 | 67.5 | 85.0 |

fusion effectively combines visual grounding and reasoning. On the more stringent MMMU-Pro, ANO-8B reaches 34.7. Increasing model capacity yields further gains: ANO-15B attains 61.3 on MMMU and 49.4 on MMMU-Pro, demonstrating that additional parameters enable richer reasoning adaptations. ANO-38B further improves to 69.2 (MMMU) and 56.8 (MMMU-Pro), outperforming InternVL2.5-38B by 5.3 and 8.8 points, respectively, and confirming a strong scaling trend.

**Math Benchmarks (MathVision, WeMath).** On MathVision, which emphasizes visual mathematical reasoning, ANO consistently achieves large improvements across scales. At 8B parameters, ANO improves from 20.1 (Idefics3) to 27.6. Scaling to 15B further boosts performance to 37.2, surpassing NVILA-15B by +14.0 points. At the large scale, ANO-38B reaches 39.7, establishing a new state-of-the-art and outperforming InternVL2.5-38B by +7.5 points. On WeMath, ANO similarly demonstrates strong reasoning gains. ANO-15B already surpasses NVILA-15B (32.3 vs. 31.1), and ANO-38B achieves 47.0, substantially outperforming InternVL2.5-38B by +8.7 points. These results highlight that our hierarchical fusion not only preserves visual grounding but also significantly strengthens symbolic reasoning, particularly in mathematically intensive tasks.

**Scale and Architecture Analysis.** Aggregating results across all three benchmarks reveals a clear scaling law: ANO variants continue to reap sustained reasoning benefits and exhibit greater robustness as model size increases. Both Qwen-based ANO-15B and LLaMA-based ANO-8B showed consistent improvements in reasoning capabilities, indicating that our layer-wise fusion is independent of the backbone architecture. Moreover, ANO-38B's strong performance on the most challenging benchmarks underscores that larger models can more fully exploit the injected reasoning capabilities, all achieved without any additional gradient-based training.

## 4.3 ABLATION STUDY

We take ANO-15B as a representative example to conduct the ablation study, since it provides a balanced scale between smaller (8B) and larger (38B) variants while sharing the same fusion principles. Table 2 reports results on MMMU, MMMU-Pro, MathVision, and WeMath using NVILA-15B and DeepSeekDistil-Qwen2.5-14B backbones without fine-tuning. Traditional baselines (VLM-Merging, AdaMMS, Task Arithmetic, and MetaGPT) yield average scores of 36.4, 36.8, 38.6,

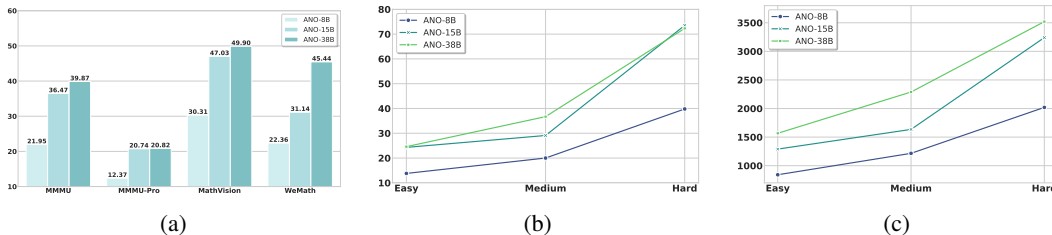

Figure 3: **(a)** Average number of reflection tokens output by ANO across different benchmarks. **(b)** Average number of reflection tokens of the ANO on the MMMU benchmark, stratified by task difficulty. **(c)** Average output length of the ANO on the MMMU benchmark, stratified by task difficulty.

and 39.6, with Task Arithmetic ($\lambda_V = 0.3$ (Ilharco et al., 2023; Zhou et al., 2024)) offering only modest gains over the baseline. **Layer-only fusion** (w/o Modality Prior) improves the average to 40.1, surpassing all traditional methods, while **ANO-15B** further incorporates the attention-guided exponential decay prior and reaches 45.1, adding +5.0 over layer-only fusion and +9.2 over baseline. This validates the role of the modality prior in balancing visual grounding and symbolic reasoning.

### 4.4 VISUAL PERCEPTION EVALUATION

We assess visual perception capability on the MME benchmark (Liang et al., 2024) (Table 3). The perceptual upper bound, corresponding to the non-reasoning multimodal model without fusion, reflects the maximum attainable accuracy on visual grounding tasks. As expected, introducing fusion into decoder layers incurs some degradation in perception. However, **ANO-15B**, equipped with our attention-guided Decay Prior, stays remarkably close to this upper bound, while substantially outperforming the ablation without the prior. In contrast, **ANO-15B w/o MP** exhibits larger drops across multiple sub-tasks, highlighting that the Modality Prior is crucial for preserving perceptual fidelity when injecting reasoning. Notably, on tasks where commonsense reasoning complements perception (e.g., "Comm."), ANO-15B even slightly exceeds the perceptual upper bound (85.0 vs. 82.9), suggesting that our fusion not only maintains but can enhance performance in scenarios where reasoning augments visual understanding. Given space constraints, we provide a more detailed discussion of this trade-off between perception and reasoning in Appendix A.9.

### 4.5 REFLECTION FREQUENCY AND OUTPUT LENGTH ANALYSIS

We analyze ANO's self-correction by counting reflection tokens (`Wait`, `Hmm`, `Mistake`, `Alternatively`, `Check`). As shown in Figure 3 (a), non-reasoning baselines output zero reflection tokens, while ANO variants consistently produce them, with larger models yielding more. This confirms that our fusion intrinsically enables iterative self-correction. Figure 3 (b) shows that reflection frequency increases monotonically with task difficulty (Easy→Hard), and larger models consistently reflect more at each level. Figure 3 (c) further reveals that output length scales with both task difficulty and model size. Together, these results demonstrate ANO's **inference scaling**: reasoning depth and reflection frequency grow adaptively with both problem complexity and model capacity. Moreover, we observe that higher reflection frequency strongly correlates with accuracy gains across benchmarks, highlighting reflection as a key driver of performance improvement.

## 5 CONCLUSIONS

In this paper, we presented ANO, a training-free and R1-like MLLM that endows off-the-shelf MLLMs with advanced reasoning and self-reflection capabilities. By decomposing the decoder into per-layer task vectors and deriving closed-form fusion weights under NTK linearization and task-vector orthogonality, ANO seamlessly integrates visual grounding from vision-fine-tuned MLLMs with logical reasoning from reasoning-specialized LLMs, guided by attention-driven modality priors. Extensive experiments demonstrate the effectiveness of our method.

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

# A    APPENDIX

## A.1    ETHICS STATEMENT

The authors have adhered to the ICLR Code of Ethics. This research did not involve any human subjects. Our work is based on pre-existing, publicly available models and datasets, and we have provided references to their original sources and licenses in Appendix A.14. The primary goal of our research is to advance the scientific understanding of reasoning capabilities in multimodal models. While we acknowledge that any powerful language model has the potential for dual use, our work does not introduce new applications that inherently carry greater risk than existing models. We believe the insights from our training-free fusion method contribute positively to the development of more transparent and interpretable AI systems.

## A.2    REPRODUCIBILITY STATEMENT

We are committed to ensuring the reproducibility of our research. Appendix A.4 provides detailed implementation specifics, including a complete breakdown of the non-reasoning MLLMs, reasoning-specialized LLMs, and base models used to construct each ANO variant (ANO-8B, ANO-15B, and ANO-38B). This section serves as a comprehensive guide for reproducing our experimental setup.

## A.3    THE USE OF LARGE LANGUAGE MODELS (LLMS)

We used a large language model (LLM) solely for minor writing assistance, such as improving grammar. The LLM was not involved in research ideation, experimental design, implementation, analysis, or interpretation of results. All scientific contributions of this paper are entirely from the authors.

## A.4    IMPLEMENTATION DETAILS

Table 4: Detailed fusion components for ANO configurations

| ANO Variant | Non-Reasoning MLLM | Reasoning LLM | Base Model |
|---|---|---|---|
| ANO-8B | Idefics3-8B | DeepSeekDistil-LLaMA3-8B | LLaMA3.1-8B |
| ANO-15B | NVILA-15B | DeepSeekDistil-Qwen2.5-14B | Qwen2.5-14B |
| ANO-38B | InternVL2.5-38B | QwQ-32B | Qwen2.5-32B |
| ANO-4B (InternVL) | InternVL3.5-4B | Qwen3-4B-Thinking | Qwen3-4B-Base |
| ANO-4B (LLaVA) | LLaVA-OneVision1.5-4B | Qwen3-4B-Thinking | Qwen3-4B-Base |
| ANO-7B (QwenVL) | QwenVL2.5-7B | OpenThought3-7B | Qwen2.5-7B-Instruction |

As shown in Table 4, we summarize the precise fusion configurations used in our ANO variant. For each model size, we list the vision-finetuned model, the reasoning-finetuned model, and the underlying base model that we linearly combine via our layer-wise Taylor-derived weights. All our experiments were conducted on NVIDIA GPUs.

In each case, the "Non-Reasoning MLLM" provides the task vector encoding of the vision-adapted decoder updates, the "Reasoning LLM" supplies the task vector for pure language reasoning, and the "Base Model" is the pre-trained backbone into which these vectors are fused according to our attention-guided exponential decay schedule.

## A.5    ROBUSTNESS OF THE MODALITY PRIOR.

We validated the robustness of the Modality Prior on three distinct MLLM families with heterogeneous vision encoders and language decoders: **ANO-8B** (SigLIP+MLP with LLaMA3.1-8B), **ANO-15B** (SigLIP+MLP with Qwen2.5-14B), and **ANO-38B** (InternViT-6B+MLP with Qwen2.5-32B). Despite these architectural differences, all three consistently exhibit exponential decay in visual attention across decoder layers, supporting the universality of the observed pattern.

We further compared alternative decay functions on MMMU. As shown in Table 5, exponential decay consistently outperforms linear and polynomial variants.

Table 5: Comparison of different decay priors on MMMU.

| Method | ANO-8B | ANO-15B | ANO-38B |
|---|---|---|---|
| Baseline (w/o prior) | 45.7 | 58.4 | 65.2 |
| Linear decay | 47.0 | 59.6 | 68.1 |
| Polynomial decay | 47.3 | 59.8 | 68.5 |
| Exponential decay (ours) | **48.3** | **61.3** | **69.2** |

These findings indicate that while modality priors are empirically motivated, the exponential formulation provides a stable and architecture-agnostic biasing scheme for training-free fusion.

## A.6 SENSITIVITY TO TASK VECTOR ORTHOGONALITY.

Our closed-form fusion relies on the assumption that task vectors from vision-centric MLLM fine-tuning and reasoning-focused LLM fine-tuning are approximately orthogonal. This is typically satisfied in practice due to the inherent modality and objective differences, as evidenced by the near-zero similarities in Figure 2.

To examine robustness when this assumption is violated, we conducted two stress tests: (1) *Shared fine-tuning data.* We fine-tuned both branches on the same textual reasoning corpus (10K samples from OpenThought-114K (Guha et al., 2025)), which increased task vector similarity to $\approx 0.1$. Fusion under this setting led to consistent performance drops across benchmarks (Table 6). (2) *Pre-blending branches.* We interpolated the MLLM and LLM parameters prior to fusion ($0.9 \times$MLLM+$0.1 \times$LLM and vice versa), raising similarity to $\approx 0.18$. Fusion again degraded performance (Table 6).

Table 6: Impact of reduced task vector orthogonality on fusion performance.

| Setting | MMMU | MMMU-Pro | MathVision | WeMath |
|---|---|---|---|---|
| ANO-8B (OpenThought-FT) | 47.5 | 34.2 | 27.4 | 11.7 |
| ANO-8B (Pre-fusion) | 44.2 | 32.9 | 22.4 | 11.4 |
| ANO-8B (Ours) | **48.3** | **34.7** | **27.6** | **11.7** |

These results confirm that performance degrades when orthogonality is weakened, highlighting its importance to our derivation.

## A.7 ON THE DESIGN AND GENERALIZABILITY OF THE ATTENTION-GUIDED DECAY PRIOR

We derived the attention-based decay prior from MSCOCO to avoid domain leakage: MSCOCO provides widely available, domain-agnostic natural images. To assess domain sensitivity, we recomputed layer-wise visual attention patterns using MMMU validation images.

Table 7: Performance of ANO-8B using modality priors derived from different datasets.

| Model | MMMU | MMMU-Pro | MathVision | WeMath |
|---|---|---|---|---|
| Idecifics3-8B (baseline) | 43.9 | 32.6 | 20.1 | 12.3 |
| ANO-8B (Ours-MMMU) | 48.5 | 34.7 | 27.1 | 11.7 |
| ANO-8B (Ours-MSCOCO) | 48.3 | 34.7 | 27.6 | 11.7 |

Table 7 shows that domain-specific priors yield only marginal gains while slightly degrading performance on other benchmarks, confirming MSCOCO as a reasonable choice. For the functional form, we adopted exponential decay due to its simplicity, interpretability, and empirical fit, while also comparing against alternative formulations to ensure robustness.

## A.8 COMPATIBILITY WITH FINE-TUNING

Our primary contribution lies in proposing a training-free fusion framework. Nonetheless, we acknowledge that downstream fine-tuning can further enhance performance in practical deployments.

To assess this, we conducted a small-scale experiment on ANO-8B using the Multimodal-Cold-Start dataset (Wei et al., 2025a), which contains 51,534 multimodal reasoning samples.

Table 8: Fine-tuning results of ANO-8B on four benchmarks.

| Model | MMMU | MMMU-Pro | MathVision | WeMath |
|---|---|---|---|---|
| Idefics3-8B (baseline) | 43.9 | 32.6 | 20.1 | 12.3 |
| Idefics3-8B (fine-tune) | 45.0 | 33.1 | 22.3 | 12.7 |
| ANO-8B (Ours) | 48.3 | 34.7 | 27.6 | 11.7 |
| ANO-8B (Ours fine-tune) | **49.2** | **35.5** | **28.9** | **12.9** |

Table 8 shows that ANO-8B (fine-tune) consistently outperforms both the baseline and the fine-tuned MLLM. This demonstrates a synergistic effect: our fusion framework provides a strong training-free initialization, while lightweight supervised fine-tuning further adapts the merged model to downstream tasks. Importantly, this highlights the flexibility of our approach, fully training-free by design, yet fully compatible with fine-tuning when practical deployment demands task-specific optimization.

## A.9 Analysis of ANO Performance on MathVista Benchmark

Tables 9 and 10 summarize both the sub-task level breakdown and overall MathVista (Lu et al., 2024) accuracy across different models. For ANO-15B, the largest degradation occurs on fine-grained visual tasks such as Bar Chart, Line Plot, and Scatter Plot (drops of 16.7-38.6% relative to NVILA-15B), where low-resolution inputs hinder precise digit recognition. In contrast, on reasoning-oriented tasks such as PaperQA, VQA-AS, and GEOS, ANO-15B outperforms NVILA-15B by 13-26%, confirming the effectiveness of reasoning transfer. This trade-off diminishes with larger models: ANO-38B not only recovers the gap but surpasses InternVL2.5-38B on MathVista (73.1 vs. 71.9), demonstrating that scaling alleviates perceptual precision loss while preserving reasoning benefits.

## A.10 Scaling ANO with Advancing MLLMs

To further assess the scalability of ANO, we applied our fusion strategy to the latest generation of MLLMs and reasoning LLMs. Table 11 reports results across four benchmarks (MMMU, MathVista, MathVision, and WeMath), as most compared models do not report MMMU-Pro, and thus we omit it here. Notably, fusing InternVL3.5-4B with Qwen3-4B yields **ANO-4B**, which achieves 69.6 on MMMU, 77.6 on MathVista, 55.1 on MathVision, and 51.9 on WeMath, surpassing its base MLLM (InternVL3.5-4B) by 3.0, 0.5, 0.7, and 1.8 points, respectively. Unlike earlier results, no performance drop is observed on MathVista. This is because the multimodal branch (InternVL3.5-4B) has already undergone reinforcement learning to acquire reasoning capabilities. Consequently, the role of LLM fusion is no longer to endow reasoning from scratch but to further strengthen it, resulting in lower fusion weights on the LLM branch and thus better preservation of perceptual capacity.

These findings validate that ANO remains effective as architectures evolve, consistently enhancing reasoning without retraining. More broadly, our framework complements advances in scale and design: while frontier MLLMs already achieve strong performance, ANO provides an orthogonal, training-free path for sustained improvements, demonstrating robustness and long-term applicability.

## A.11 Validation of Layer-Wise Multimodal Prior and Task-Vector Orthogonality

To verify the two key assumptions underpinning our fusion strategy, namely that (1) shallow decoder layers focus predominantly on visual perception while deeper layers prioritize language reasoning (the multimodal prior), and (2) task vectors from vision-finetuned and reasoning-finetuned models are mutually orthogonal, we analyze both Idefics3-8B and NVILA-15B in Figures 4 and 5.

**Layer-Wise Attention Patterns** Figures 4 (a) and 5 (a) plot the average per-layer attention weights that each model's decoder assigns to visual tokens when processing vision inputs. In both the 8B and 38B variants, we observe a clear decay: the first few layers exhibit high visual attention, and then

Table 9: Sub-task analysis of MathVista (ANO-15B vs. NVILA-15B).

| Task | NVILA-15B | ANO-15B | Improvement |
|------|-----------|---------|-------------|
| Bar Chart | 79.8 | 41.2 | -38.6% |
| Line Plot | 74.4 | 48.7 | -25.7% |
| Scatter Plot | 66.7 | 50.0 | -16.7% |
| PaperQA | 57.9 | 84.2 | +26.3% |
| VQA-AS | 71.4 | 85.7 | +14.3% |
| GEOS | 40.9 | 54.5 | +13.6% |

Table 10: MathVista accuracy across different models.

| Model | Accuracy (%) |
|-------|-------------|
| Idefics3-8B | 58.4 |
| ANO-8B | 50.7 |
| NVILA-15B | 67.6 |
| ANO-15B | 55.4 |
| InternVL2.5-38B | 71.9 |
| ANO-38B | 73.1 |

Table 11: Performance of recent MLLMs and LLMs on four benchmarks. ANO-4B is obtained by fusing InternVL3.5-4B with Qwen3-4B. Our method continues to provide consistent improvements as MLLMs advance.

| Model | MMMU | MathVista | MathVision | WeMath |
|-------|------|-----------|------------|--------|
| Gemma3-4B (Team et al., 2025a) | 47.3 | 46.3 | 23.8 | 26.7 |
| LLaVA-OV-7B (Li et al., 2024) | 47.9 | 58.6 | 18.3 | 20.9 |
| InternVL2.5-8B (Chen et al., 2024c) | 56.2 | 64.5 | 17.0 | 23.5 |
| Qwen2.5-VL-8B (Bai et al., 2025) | 55.0 | 67.8 | 25.4 | 35.2 |
| Gemma3-12B (Team et al., 2025a) | 55.2 | 56.1 | 30.7 | 33.6 |
| Gemma3-27B (Team et al., 2025a) | 64.8 | 59.8 | 36.4 | 40.7 |
| Qwen2.5-VL-72B (Bai et al., 2025) | 68.2 | 74.2 | 39.3 | 49.1 |
| InternVL2.5-78B (Chen et al., 2024c) | 70.0 | 72.3 | 32.2 | 39.8 |
| Claude-3.7-Sonnet (Claude, 2024) | 75.0 | 66.8 | 41.9 | 49.3 |
| Gemini-2.0-Flash (Gemini 2.0, 2024) | 72.6 | 70.4 | 43.6 | 47.4 |
| Gemini-2.0-Pro (Gemini 2.0, 2024) | 69.9 | 71.3 | 48.1 | 56.9 |
| InternVL3.5-4B (Wang et al., 2025) (Baseline) | 66.6 | 77.1 | 54.4 | 50.1 |
| **ANO-4B (Ours)** | 69.6 | 77.6 | 55.1 | 51.9 |

steadily decline toward deeper layers. This consistent decay profile confirms our multimodal prior hypothesis across both scales, justifying the use of an exponential decay schedule guided by these attention statistics.

**Task-Vector Cosine Similarity** Figures 4 (b) and 5 (b) report the cosine similarity between the task vectors extracted from vision-finetuned models (Idefics3-8B / NVILA-15B) and those from reasoning-finetuned counterparts (DeepSeekDistil-LLaMA3-8B / DeepSeekDistil-Qwen2.5-14B) at each decoder block. In both cases, the similarity values remain close to zero across almost all layers, confirming that the vision and reasoning task vectors occupy nearly orthogonal subspaces. This orthogonality underpins our linear fusion derivation via Taylor expansion, ensuring that the combined update remains a meaningful superposition of the two modalities without destructive interference.

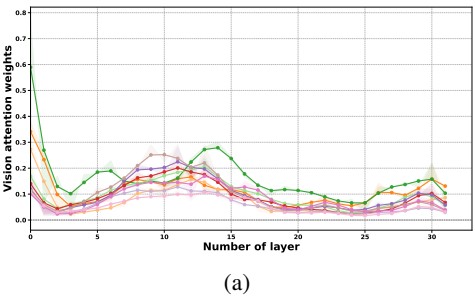
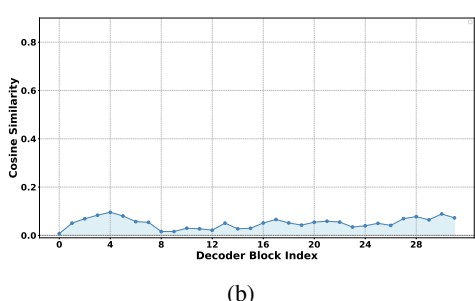

(a)  (b)

Figure 4: **(a)** Layer-wise visual attention of Idefics3-8B. **(b)** Cosine similarity between task vectors of vision-finetuned (Idefics3-8B) and reasoning-finetuned (DeepSeekDistil-LLaMA3-8B) models at each decoder block.

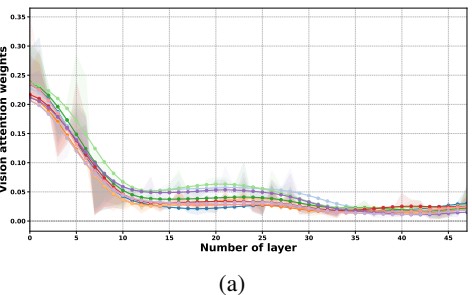 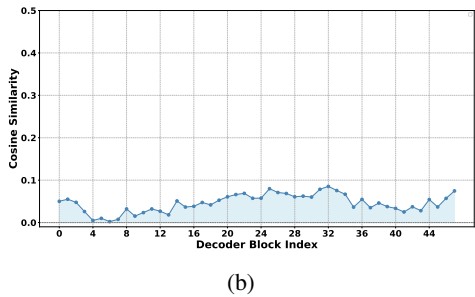

(a)  (b)

Figure 5: **(a)** Layer-wise visual attention of NVILA-15B. **(b)** Cosine similarity between task vectors of vision-finetuned (NVILA-15B) and reasoning-finetuned (DeepSeekDistil-Qwen2.5-14B) models at each decoder block.

### A.12 CASE STUDY: EXEMPLARY MODEL OUTPUTS

To illustrate the qualitative improvements brought by our ANO fusion, we present three representative examples, one for each model size (8B, 15B, 38B), comparing a non-reasoning baseline against the corresponding ANO variant. In each case, the baseline model provides a direct (and incorrect) answer, whereas the ANO model produces a detailed chain-of-thought with reflection tokens, arriving at the correct solution.

As shown in Figures 6, Figures 7, and Figures 8, across all three model scales, the ANO variants consistently produce step-by-step reasoning chains and explicit reflection checks. This enables them to correct intermediate arithmetic or logical reasoning and arrive at the correct results. In contrast, the non-reasoning baselines directly output an incorrect answer without any justification.

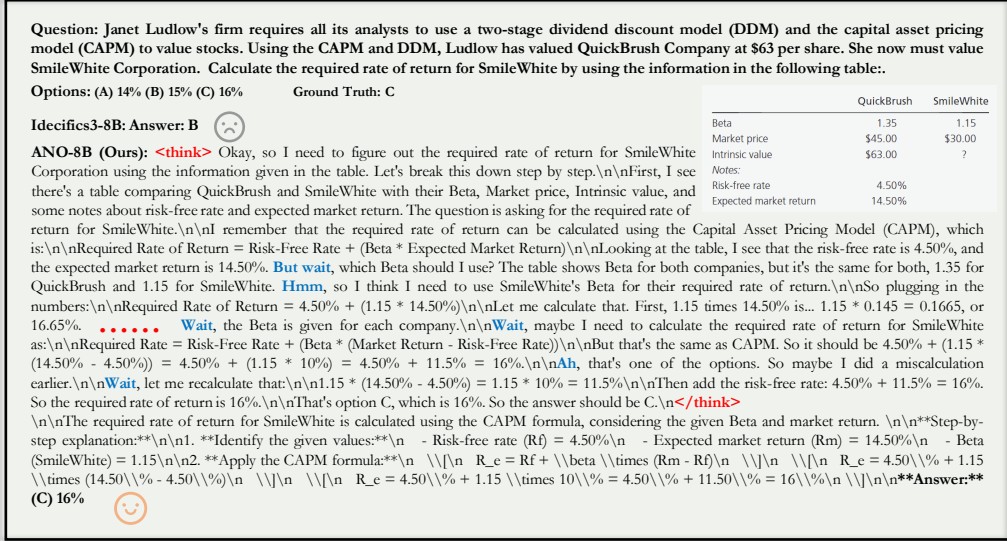

Figure 6: Output examples from ANO-8B and the non-reasoning baseline model Idecifics3-8B. Here, ⟨ think ⟩ and ⟨ /think ⟩ denote R1-like reasoning processes, while blue text indicates reflection tokens.

### A.13 LIMITATIONS AND FUTURE DIRECTIONS

While ANO demonstrates strong performance on vision-language reasoning, its current formulation is limited to vision inputs and text-based reasoning modules. In particular, our multimodal prior and task-vector fusion have only been validated on image-text pairs. Modalities such as audio, point

**The jet engine in Fig. P3.50 admits air at 20\u00b0C and 1 atm at (1), where A_1 = 0.5 m^2 and V_1 = 250 m/s. The fuel-air ratio is 1:30. The air leaves section (2) at 1 atm, V_2 = 900 m/s, and A_2 = 0.4 m^2. Compute the test stand support reaction R_x needed.**
**Options: (A) 99,000 N (B) 102,000 N (C) 121,000 N        Ground Truth: B**

**NVIL-15B: (C)** ☹

**ANO-15B (Ours): <think>** Okay, so I have this problem about a jet engine, and I need to find the test stand support reaction R_x. Hmm, let me try to visualize the setup based on the description. There's a combustion chamber with air entering at point 1 and exiting at point 2. The engine is on a test stand, so it's probably stationary, and we need to calculate the force it exerts on the stand.\n\nFirst , I remember that in fluid dynamics, when dealing with forces on a control volume, we can use the momentum equation. The general form is \u03a3F = m_dot*(V2 - V1) + (pA)(V2 - V1) terms, but since the areas are different, maybe it's better to use the integral form of the momentum equation.\n\n**Wait**, the problem gives me the areas A1 and A2, and the velocities V1 and V2. Also, the fuel-air ratio is 1:30, which I think means that for every 30 units of air, there's 1 unit of fuel. But since fuel is burned in the combustion chamber, the mass flow rate of the air will increase because fuel is added. **Hmm**, but I'm not sure if I need to consider the fuel mass flow here. Let me think.\n\nThe support reaction R_x is the force that the engine exerts on the test stand. By Newton's third law, the reaction force would be equal and opposite to the thrust force produced by the engine. So, if I can find the thrust, that would give me R_x.\n\nThrust in a jet engine is typically calculated as the mass flow rate of the air multiplied by the velocity of the exhaust gases minus the incoming air velocity. **But wait**, in this case, the engine is on a test stand, so maybe the thrust is just the difference in momentum flow rates at the inlet and outlet.\n\nLet me write down the given data:\n\nAt inlet (1):\n- Temperature T1 = 20\u00b0C, which is 293.15 K.\n- Pressure P1 = 1 atm.\n- Area A1 = 0.5 m\u00b2.\n- Velocity V1 = 250 m/s.\n\nAt outlet (2):\n- Pressure P2 = 1 atm.\n- Velocity V2 = 900 m/s.\n- Area A2 = 0.4 m\u00b2.\n\nFuel-air ratio is 1:30, so for every 30 kg of air, 1 kg of fuel is burned. So, the total mass flow rate at the exit would be the mass flow rate of air plus the mass flow rate of fuel.\n\n**But wait**, the mass flow rate of air at the inlet is m_dot_air = rho1 * A1 * V1. Then, the mass flow rate of fuel is m_dot_fuel = m_dot_air / 30. So, the total mass flow rate at the exit is m_dot_total = m_dot_air + m_dot_fuel = m_dot_air * (1 + 1/30) = m_dot_air * (31/30).\n **• • • • • •** **Wait**, but the pressure at both inlet and outlet is 1 atm, so the pressure forces would be p*A1*V1 and p*A2*V2, but I'm not sure if they contribute to the net force. **Alternatively**, maybe I should consider the momentum equation as:\n\nR_x = m_dot2 * V2 - m_dot1 * V1.\n\n **But wait**, that's what I did earlier, and I got approximately 102,234 N, which is close to option B, 102,000 N.\n\nAlternatively, maybe I should consider the specific heat or the energy balance, but I think the momentum equation approach is sufficient here.\n\n**Wait**, another thought: The mass flow rate at the exit includes both the incoming air and the fuel. So, m_dot2 = m_dot_air + m_dot_fuel = m_dot_air + (m_dot_air / 30) = (31/30)*m_dot_air.\n\nWhich is what I did earlier. So, I think my calculation is correct.\n\nSo, the support reaction R_x is equal to the thrust, which is approximately 102,234 N, so the closest option is B) 102,000 N.\n**</think>**
\n\nTo determine the test stand support reaction \\( R_x \\), we apply the momentum equation. The thrust \\( T \\) is calculated as the difference in momentum flow rates at the inlet and outlet:\n\n\\[\nT = \\dot{m}_2 V_2 - \\dot{m}_1 V_1\n\\]\n\nGiven the fuel-air ratio of 1:30, the total mass flow rate at the exit is \\( \\dot{m}_2 = \\frac{31}{30} \\dot{m}_1 \\). Using the ideal gas law, the density \\( \\rho_1 \\) at the inlet is calculated, and the mass flow rates \\( \\dot{m}_1 \\) and \\( \\dot{m}_2 \\) are determined. Substituting these into the thrust equation yields \\( T \\approx 102,234 \\, \\text{N} \\), which rounds to the closest option:\n\n**Answer: (B) 102,000 N** ☺

Figure 7: Output examples from ANO-15B and the non-reasoning baseline model NVILA-15B.

**Which reaction in the following reaction sequence <image 1> is regioselective?**
**Ground Truth: B**

**InternVL2.5-38B: Step A** ☹

**ANO-38B (Ours): <think>** Okay, let me try to work through this problem step by step. The question is asking which reaction in the given reaction sequence is regioselective. The image shows a series of four steps labeled A, B, C, D. I need to figure out which of these steps involves regioselectivity.\n\nFirst, I should recall what regioselective means. reaction is one where the major product forms in a specific region of the molecule, favoring one structural form over another possible isomer. This is different from stereoselectivity, which is about the spatial arrangement.\n\nLooking at the reaction steps in the image:\n\nStep A: The reactants start with a compound that has an OMs (maybe mesylate?) group. The reagent used here is NaHCO3. **Wait**, the image might be a bit hard to interpret, but the reaction might be an acid-base reaction or maybe deprotonation. **Alternatively**, if it's a substitution reaction, like maybe an esterification? But with NaHCO3? **Hmm**. Let me sketch the structure based on the details given.\n\nStep A shows a molecule with an OMS group attached to a benzene ring. When treated with NaHCO3 (which is a weak base), perhaps it results in deprotonation somewhere. But the product after Step A has an ester group (COOEt) attached. Maybe in Step A, there's a substitution where OMs leaves, and EtCO-comes in? **Wait**, but sodium bicarbonate might be used in some way to deprotonate and form a carboxylate.\n\n **Alternatively**, perhaps Step A is an saponification or some ester hydrolysis? The OMS group might be a leaving group, and the base (NaHCO3) could facilitate substitution of that leaving group with something else. But I'm a bit confused here.\n\n**Wait**, maybe the first step is a Swern oxidation? No, the reagents are NaHCO3. **Wait**, looking at the structure after Step A, maybe there's an ester group on the same benzene ring as where OMs was. Did the OMS get replaced by an ester group? If so, perhaps it's nucleophilic substitution at the benzylic position?\n **• • • • • •** \n\n **Alternatively**, maybe the Staudinger reaction in Step C produces an amine in a certain position, but that's part of the reduction step. Since reduction of azide to amine doesn't alter the position, it's not regioselective there.\n\nTherefore, I think the regioselective step is Step B. Because in aromatic substitution with sodium azide, the position where azide adds depends on the directing groups present on the molecule. So if there are activating groups like OH or other substituents directing the azide to a specific position (ortho or para to those groups), then Step B is regioselective.\n**</think>**
\n\nThe reaction that is regioselective is **Step B**. Here's the analysis:\n\n### Key Observations:\n\n1. **Step A**: The displacement of the mesylate (OMs) by an ethyl ester (COEt) occurs at the same position where the mesylate (OMs) was located. Since the nucleophile (EtCO\u207b) attacks the carbon bearing the leaving group, this step is not regioselective as it does not depend on existing directing groups.\n\n2. **Step B**: This involves introducing an azide group (N\u2083\u207b) via nucleophilic aromatic substitution. The position where azide attacks will be directed by activating or deactivating groups already present on the benzene ring. For example, if the benzene ring has an electron-donating group like a hydroxyl (\u2013OH), it will direct the azide to the **ortho** or **para** positions relative to \u2013OH. This directed substitution introduces regioselectivity.\n\n3. **Step C**: Reduction of azide to amine (via PPh\u2083 / NEt\u2083) does not alter regiochemistry\u2014the position of the new amine group is fixed from the prior substitution (Step B).\n\n4. **Step D**: This step modifies an amine group (N) with methanesulfonyl chloride, producing an N-sulfonyl derivative. The substitution occurs at the nitrogen (not the aromatic ring), so this is not regioselective in an aromatic sense.\n\n### Conclusion:\n\n**Step B** is the only step where the regiochemistry is influenced by existing substituents (e.g., activating groups like \u2013OH) on the aromatic ring, making it regioselective.\n\n**Answer: **Step B** is regioselective.** ☺

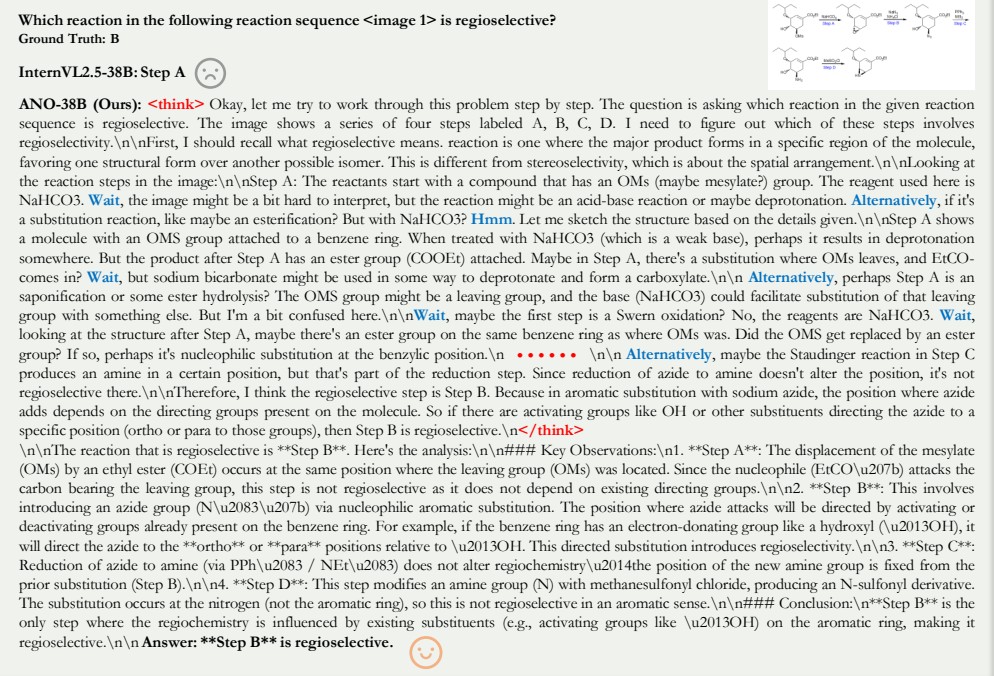

Figure 8: Output examples from ANO-38B and the non-reasoning baseline model InternVL2.5-38B.

clouds, or structured data exhibit very different layer-wise attention and feature dynamics, and may require distinct prior shapes or alternative fusion mechanisms.

## A.14 LICENSES FOR EXISTING ASSETS

During our experiments, we utilized publicly available models and datasets. Specifically, Idecifics3-8B[1], NVILA-15B[2], InternVL2.5-38B[3], DeepSeekDistil-LLaMA3-8B[4], DeepSeekDistil-Qwen2.5-14B[5], QwQ-32B[6], LLaMA3.1-8B[7], Qwen2.5-14B[8], and Qwen2.5-32B[9] were used under proprietary licenses. Similarly, the datasets used in our experiments were made publicly accessible by their original authors through the following links: MMMU[10], MMMU-Pro[11], MathVista[12], MathVision[13], Wemath[14], and MME[15]. In our paper, we have cited and introduced each comparative method and dataset accordingly.

## A.15 PROOF OF NTK LINEARIZATION (PROPERTY)

Under the Neural Tangent Kernel (NTK) regime for wide transformers, the network behavior near initialization $\theta_0$ admits:

$$f(x; \theta_0 + \Delta\theta) = f(x; \theta_0) + \nabla_\theta f(x; \theta_0)^\top \Delta\theta + \mathcal{O}(\|\Delta\theta\|^2 / \sqrt{\text{width}}). \tag{17}$$

For task-specific loss $\mathcal{L}_t^{(l)}$, the Hessian exhibits two key properties: **Jacobian dominance**: Residual terms vanish due to $\nabla^2 f^{(l)} \sim \mathcal{O}(1/\sqrt{\text{width}})$. **Isotropy**: Gradient directions become nearly orthogonal in high dimensions. This leads to the layer-wise Hessian approximation:

$$\nabla^2 \mathcal{L}_t^{(l)}(\theta) \approx \mathbb{E}_x \left[ \nabla f^{(l)} \nabla f^{(l)\top} \right] \approx \delta_t^{(l)} I_{d_l}, \tag{18}$$

where the curvature scalar:

$$\delta_t^{(l)} = \frac{1}{d_l} \text{Tr} \left( \nabla^2 \mathcal{L}_t^{(l)}(\theta_0^{(l)}) \right) = \frac{1}{d_l} \mathbb{E}_x \left[ \|\nabla f^{(l)}(x; \theta_0^{(l)})\|^2 \right], \tag{19}$$

captures the average gradient magnitude at initialization.

## A.16 PROOF OF TASK-VECTOR ORTHOGONALITY (PROPERTY)

Figure 2(a) shows that, vision-fine-tuning and reasoning-fine-tuning update disjoint aspects of the LLM's representations. Concretely, let

$$\tau_V^{(l)} = \theta_V^{(l)} - \theta_0^{(l)}, \quad \tau_R^{(l)} = \theta_R^{(l)} - \theta_0^{(l)}. \tag{20}$$

We observe that

$$(\tau_V^{(l)})^\top \tau_R^{(l)} = \mathcal{O}(\varepsilon), \tag{21}$$

where $\varepsilon$ is a small constant, indicating near-orthogonality.

---

[1] https://huggingface.co/HuggingFaceM4/Idefics3-8B-Llama3.
[2] https://huggingface.co/Efficient-Large-Model/NVILA-15B
[3] https://huggingface.co/OpenGVLab/InternVL2_5-38B
[4] https://huggingface.co/deepseek-ai/DeepSeek-R1-Distill-Llama-8B
[5] https://huggingface.co/deepseek-ai/DeepSeek-R1-Distill-Qwen-14B
[6] https://huggingface.co/Qwen/QwQ-32B
[7] https://huggingface.co/meta-llama/Llama-3.1-8B
[8] https://huggingface.co/Qwen/Qwen2.5-14B
[9] https://huggingface.co/Qwen/Qwen2.5-32B.
[10] https://huggingface.co/datasets/MMMU/MMMU.
[11] https://huggingface.co/datasets/MMMU/MMMU_Pro.
[12] https://huggingface.co/datasets/AI4Math/MathVista.
[13] https://huggingface.co/datasets/MathLLMs/MathVision
[14] https://huggingface.co/datasets/We-Math/We-Math
[15] https://huggingface.co/datasets/darkyarding/MME/tree/main.

### A.17 Proof of Taylor Expansion of Layer Task Loss Difference (Lemma)

To quantify the impact of parameter fusion on task performance, we analyze the Layer-wise Task Loss Difference (LTLD) through a rigorous Taylor expansion approach. The fused parameters combine both task updates through convex combination:

$$\theta_f^{(l)} = \theta_0^{(l)} + \sum_{t \in \{V,R\}} \lambda_t^{(l)} \tau_t^{(l)}, \quad \text{where } \lambda_t^{(l)} \in [0, 1], \tag{22}$$

where, $\theta_0^{(l)}$ denotes the pre-trained initialization at layer $l$, $\theta_t^{(l)} = \theta_0^{(l)} + \tau_t^{(l)}$ denotes the fine-tuned parameters for task $t \in \{V, R\}$, $\tau_t^{(l)}$ denotes the task-vector update from $\theta_0^{(l)}$ to $\theta_t^{(l)}$.

**Fusion Residual Vector**: Measures deviation from optimal task parameters:

$$h_t^{(l)} = \theta_f^{(l)} - \theta_t^{(l)} = \sum_{k \neq t \in \{V,R\}} \lambda_k^{(l)} \tau_k^{(l)} - (1 - \lambda_t^{(l)}) \tau_t^{(l)}. \tag{23}$$

**Interpolation Path**: Defines a linear trajectory in parameter space from the fine-tuned parameters $\theta_t^{(l)}$ to the fused parameters $\theta_f^{(l)}$, enabling exact Taylor expansion along the fusion direction:

$$\gamma_t^{(l)}(\beta) = \theta_t^{(l)} + \beta h_t^{(l)}, \quad \beta \in [0, 1]. \tag{24}$$

**Taylor Expansion Analysis:** Applying second-order expansion along $\gamma_t^{(l)}$:

$$\begin{aligned}
\mathcal{L}_t^{(l)}(\theta_f^{(l)}, x_t) &= \mathcal{L}_t^{(l)}(\theta_t^{(l)} + h_t^{(l)}, x_t) \\
&= \mathcal{L}_t^{(l)}(\theta_t^{(l)}, x_t) + \nabla \mathcal{L}_t^{(l)}(\theta_t^{(l)}, x_t)^\top h_t^{(l)} \\
&\quad + \frac{1}{2} h_t^{(l)\top} \left( \int_0^1 \nabla^2 \mathcal{L}_t^{(l)}(\gamma_t^{(l)}(\beta)) d\beta \right) h_t^{(l)}
\end{aligned} \tag{25}$$

Under fine-tuning convergence $\nabla \mathcal{L}_t^{(l)}(\theta_t^{(l)}, x_t) \approx 0$, we obtain:

$$\begin{aligned}
\text{LTLD}_t^{(l)} &= \mathcal{L}_t^{(l)}(\theta_f^{(l)}, x_t) - \mathcal{L}_t^{(l)}(\theta_t^{(l)}, x_t) \\
&= \frac{1}{2} h_t^{(l)\top} \left( \int_0^1 \nabla^2 \mathcal{L}_t^{(l)}(\gamma_t^{(l)}(\beta)) d\beta \right) h_t^{(l)}.
\end{aligned} \tag{26}$$

This derivation establishes the theoretical foundation for our layer-wise fusion analysis, connecting parameter perturbations to task performance through differentiable geometry.

### A.18 Derivation of Closed-Form Fusion Weights

To enable efficient model merging while preserving task performance, we derive theoretically-grounded fusion weights through layer-wise NTK and Task-Vector Orthogonality analysis. The key insight is that LLMs exhibit approximately quadratic loss landscapes under NTK, permitting closed-form solutions. Under the NTK linearization regime for LLMs, we first analyze the layer-wise behavior. For any layer $l$ with initialization $\theta_0^{(l)}$ and perturbation $\tau^{(l)} = \theta^{(l)} - \theta_0^{(l)}$, the parameter admits the first-order approximation:

$$f^{(l)}(x; \theta^{(l)}) \approx f^{(l)}(x; \theta_0^{(l)}) + \underbrace{\nabla_{\theta^{(l)}} f^{(l)}(x; \theta_0^{(l)})^\top \tau^{(l)}}_{\text{Linear term}} + \mathcal{O}(\|\tau^{(l)}\|^2) \tag{27}$$

This linearity emerges in wide transformers where the network's output changes linearly with parameter perturbations.

For the quadratic loss $\mathcal{L}_t^{(l)}(\theta^{(l)}, x_t) = \frac{1}{2} \| f^{(l)}(x_t; \theta^{(l)}) - y_t \|^2$, we compute the Hessian:

$$\begin{aligned}
\nabla^2 \mathcal{L}_t^{(l)} &= \nabla f^{(l)}(x_t; \theta^{(l)}) \nabla f^{(l)}(x_t; \theta^{(l)})^\top \\
&\quad + \underbrace{(f^{(l)}(x_t; \theta^{(l)}) - y_t)^\top \nabla^2 f^{(l)}(x_t; \theta^{(l)})}_{\text{Vanishes under NTK regime}}
\end{aligned} \tag{28}$$

Under the NTK regime, the second term becomes negligible due to two intrinsic properties of wide neural networks: 1) the output residual $\|f^{(l)} - y_t\|$ vanishes with near-optimal fine-tuning (guaranteed by NTK's convex-like optimization landscape), and 2) the Hessian $\nabla^2 f^{(l)}$ shrinks as $\mathcal{O}(1/\sqrt{\text{width}})$ (a direct consequence of NTK's linearization effect).

Under NTK conditions:

$$\nabla^2 \mathcal{L}_t^{(l)}(\theta^{(l)}) \approx \nabla f^{(l)}(x_t; \theta_0^{(l)})\nabla f^{(l)}(x_t; \theta_0^{(l)})^\top \tag{29}$$

Taking the isotropic approximation for layers:

$$\nabla^2 \mathcal{L}_t^{(l)} \approx \delta_t^{(l)} I_{d_l}, \quad \delta_t^{(l)} = \frac{1}{d_l}\text{tr}\left(\nabla f^{(l)}(x_t; \theta_0^{(l)})\nabla f^{(l)}(x_t; \theta_0^{(l)})^\top\right) \tag{30}$$

where $d_l$ is the parameter dimension at layer $l$. This follows from the observation that gradient directions become nearly orthogonal in high dimensions.

Substituting the Hessian approximation into the Taylor remainder:

$$\int_0^1 \nabla^2 \mathcal{L}_t^{(l)}(\gamma_t^{(l)}(\beta))\, d\beta \approx \delta_t^{(l)} I, \tag{31}$$

collapses the quadratic form to a scalar multiple of $\|h_t^{(l)}\|^2$, yielding a tractable bound and closed-form solution for fusion weights. The second-order Taylor remainder is bounded by the extremal eigenvalues of the integral Hessian. Substituting our isotropic approximation:

$$2\text{LTLD}_t^{(l)} \leq h_t^{(l)\top}\left(\delta_t^{(l)} I_{d_l}\right)h_t^{(l)}$$

$$= \delta_t^{(l)}\|h_t^{(l)}\|^2$$

$$= \delta_t^{(l)}\left\|\sum_{k\neq t\in\{V,R\}}\lambda_k^{(l)}\tau_k^{(l)} - (1-\lambda_t^{(l)})\tau_t^{(l)}\right\|^2 \tag{32}$$

where we used the identity $\|a+b\|^2 = \|a\|^2 + \|b\|^2 + 2a^\top b$. The cross-term vanishes due to task vector orthogonality. This demonstrates that the layer-wise task loss difference is governed by the product of the initialization curvature $\delta_t^{(l)}$ and the squared fusion residual norm $\|h_t^{(l)}\|^2$, justifying our layer-wise analysis.

Next, we show the optimal layer-wise fusion weights. The layer-wise analysis enables independent optimization at each layer. The LALD decomposes into layer-specific terms:

$$\text{LALD}^{(l)} \leq \sum_{t\in\{V,R\}}\frac{\delta_t^{(l)}}{2}\|\tau_t^{(l)}\|^2\left[(1-\lambda_t^{(l)})^2\|\tau_t^{(l)}\|^2 + \sum_{k\neq t\in\{V,R\}}(\lambda_k^{(l)})^2\|\tau_k^{(l)}\|^2\right] \tag{33}$$

This additive structure permits layer-wise optimization by solving $L$ (where $L$ is the total number of layers) independent problems:

$$\{\lambda_t^{(l)}\}_{t\in\{V,R\}} = \arg\min J^{(l)}, \quad \forall l = 1, ..., L \tag{34}$$

Focusing on a single layer (omitting superscript $(l)$ for clarity), we reformulate the objective:

$$J = \sum_{t\in\{V,R\}}\frac{\delta_t}{2}\|\tau_t\|^2\left[(1-\lambda_t)^2\|\tau_t\|^2 + \sum_{k\neq t\in\{V,R\}}\lambda_k^2\|\tau_k\|^2\right]$$

$$= \frac{1}{2}\sum_{t\in\{V,R\}}\delta_t\|\tau_t\|^4(1-\lambda_t)^2 + \frac{1}{2}\sum_{t\in\{V,R\}}\sum_{k\neq t\in\{V,R\}}\delta_t\lambda_k^2\|\tau_t\|^2\|\tau_k\|^2 \tag{35}$$

Under task orthogonality $\tau_V^\top \tau_R = 0$, the cross-derivative terms vanish, simplifying the gradient to:

$$\frac{\partial J}{\partial \lambda_t} = -\delta_t\|\tau_t\|^4(1-\lambda_t) + \lambda_t\|\tau_t\|^2\sum_{k\neq t\in\{V,R\}}\delta_k\|\tau_k\|^2$$

$$0 = -\delta_t\|\tau_t\|^4 + \lambda_t\left[\delta_t\|\tau_t\|^4 + \|\tau_t\|^2\sum_{k\neq t\in\{V,R\}}\delta_k\|\tau_k\|^2\right] \tag{36}$$

Solving the linear system yields:

$$\lambda_t = \frac{\delta_t \|\tau_t\|^4}{\delta_t \|\tau_t\|^4 + \|\tau_t\|^2 \sum_{k \neq t} \delta_k \|\tau_k\|^2} = \frac{\delta_t \|\tau_t\|^2}{\sum_{k \in \{V,R\}} \delta_k \|\tau_k\|^2} \tag{37}$$

Under the uniform curvature assumption $\delta_k \approx \delta_0$ (arising from NTK's layer-wise gradient statistics in wide networks, where $\frac{1}{d_l} \mathrm{Tr}(\nabla f^{(l)} \nabla f^{(l)\top})$ becomes task-invariant as $d_l \to \infty$), the solution simplifies to:

$$\lambda_t^{(l)} = \frac{\|\tau_t^{(l)}\|^2}{\sum_{k \in \{V,R\}} \|\tau_k^{(l)}\|^2} \quad \Rightarrow \quad \lambda_t^{(l)} \propto \|\tau_t^{(l)}\|^2 \tag{38}$$

This closed-form solution adaptively suppresses interfering task vectors while preserving target task information at each layer.

### A.19 Derivation of Attention-Guided Exponential Decay Priors

In Section 3.2.3 we introduce an attention-guided exponential schedule for modality priors. Here, we provide a step-by-step derivation.

**Modeling the decay.** We observe the layer-wise visual-attention ratio

$$a_l = \frac{1}{N} \sum_{x,h} \frac{\mathrm{Attn}_{l \to \mathrm{vis}}(x,h)}{\mathrm{Attn}_{l \to \mathrm{vis}}(x,h) + \mathrm{Attn}_{l \to \mathrm{text}}(x,h)} \quad \forall l = 1, \ldots, L, \tag{39}$$

where $x$ indexes inputs, $h$ indexes attention heads, and $N$ is a normalization factor. We posit an exponential decay model:

$$a_l \approx C \, e^{-\alpha \, l}, \tag{40}$$

where $C > 0$ and $\alpha > 0$ are unknown constants.

**Logarithmic linearization.** Taking natural logarithm on both sides of Eq. ( 40), we obtain

$$\ln a_l \approx \ln C - \alpha \, l. \tag{41}$$

Define $y_l = \ln a_l$, $x_l = l$, $b_0 = \ln C$, $b_1 = -\alpha$. Then Eq. ( 41) becomes a linear regression problem:

$$y_l \approx b_0 + b_1 \, x_l. \tag{42}$$

**Least-squares estimation.** We collect the dataset $\{(x_l, y_l)\}_{l=1}^L$ and solve the normal equations for the least-squares fit:

$$b_1 = \frac{\sum_{l=1}^L (x_l - \bar{x})(y_l - \bar{y})}{\sum_{l=1}^L (x_l - \bar{x})^2},$$
$$b_0 = \bar{y} - b_1 \, \bar{x}, \tag{43}$$

where $\bar{x} = \frac{1}{L} \sum_l x_l$ and $\bar{y} = \frac{1}{L} \sum_l y_l$. We then recover

$$\hat{\alpha} = -b_1, \quad \hat{C} = e^{b_0}. \tag{44}$$

**Constructing modality priors.** Finally, we define the normalized exponential priors

$$w_V^{(l)} = \frac{e^{-\hat{\alpha} \, l}}{\sum_{j=1}^L e^{-\hat{\alpha} \, j}}, \quad w_R^{(l)} = 1 - w_V^{(l)}. \tag{45}$$

These priors smoothly interpolate from strong visual emphasis in early layers to strong reasoning emphasis in late layers, and can be replaced by alternative data-driven schedules if desired.

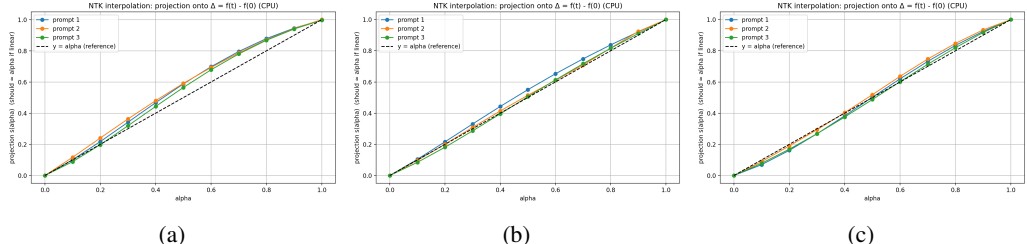

(a)                (b)                (c)

Figure 9: **NTK-style interpolation diagnostics on three multimodal decoders.** For each model, (a) Idefics3-Llama-8B, (b) NVILA-15B, and (c) InternVL2.5-38B, we linearly interpolate parameters $\theta(\alpha) = (1 - \alpha)\theta_0 + \alpha\theta_t$, evaluate logits on several prompts, and project $(f(\theta(\alpha)) - f(\theta_0))$ onto the endpoint displacement $\Delta = f(\theta_t) - f(\theta_0)$. Under NTK-style linearization, the projection should satisfy $s(\alpha) \approx \alpha$. Across all models and prompts, the curves closely follow the reference line $y = \alpha$, indicating strong empirical linearity and suggesting that larger MLLMs operate deeper in the NTK regime.

### A.20 NTK-STYLE LINEARIZATION DIAGNOSTICS

To empirically assess the validity of the NTK-style linear regime assumed in our second-order Taylor derivation, we conduct a direct interpolation diagnostic on three representative multimodal decoders: **Idefics3-Llama 8B**, **NVILA 15B**, and **InternVL2.5 38B**. For each model pair $(\theta_0, \theta_t)$, we linearly interpolate parameters

$$\theta(\alpha) = (1 - \alpha)\theta_0 + \alpha\theta_t, \qquad \alpha \in [0, 1], \tag{46}$$

evaluate the flattened logits $f(\theta(\alpha))$ on several prompts, and project the displacement $(f(\theta(\alpha)) - f(\theta_0))$ onto the endpoint direction $\Delta = f(\theta_t) - f(\theta_0)$.

Formally, the projection coefficient is

$$s(\alpha) = \frac{(f(\theta(\alpha)) - f(\theta_0))^\top \Delta}{\|\Delta\|^2}, \tag{47}$$

which should satisfy $s(\alpha) \approx \alpha$ if the model behaves linearly under NTK-style interpolation. We additionally compute the normalized residual

$$\rho(\alpha) = \frac{\|f(\theta(\alpha)) - f(\theta_0) - \alpha\Delta\|}{\|\Delta\|}, \tag{48}$$

which directly measures the magnitude of nonlinear deviation.

As shown in Figure 9, across all prompts and all three model scales, the projection curves $s(\alpha)$ closely track the ideal reference line $y = \alpha$, and the residuals remain consistently small. Larger models (e.g., InternVL2.5-38B) exhibit even tighter adherence, indicating that multimodal decoders increasingly operate in the NTK-like linear regime at scale. These results provide concrete, quantitative validation for the linearization assumptions underpinning our closed-form fusion rule.

### A.21 COMPARISON WITH DOGE

ANO and DOGE share a similar theoretical starting point: both build a data-free merging objective from a local Taylor expansion with NTK-style linearization and assume approximate orthogonality among task vectors. However, their goals and applicability diverge significantly. DOGE is designed for multi-task merging in unimodal LLMs, focusing on resolving conflicts between multiple expert task vectors. In contrast, ANO targets cross-modal capability injection, leveraging the structured layer specialization in multimodal decoders (shallow layers for perception and deeper layers for reasoning), which DOGE does not model.

To assess whether DOGE can transfer to heterogeneous multimodal architectures, we applied DOGE to InternVL3.5, the same backbone used for ANO-4B, and evaluated on MMMU. Results are shown in Table 12. DOGE severely degrades performance, whereas ANO yields a substantial improvement. This mismatch indicates that DOGE's assumptions, suitable for unimodal LLM merging, break down

in multimodal settings where cross-modal interactions and modality-specific functional structure must be explicitly considered. ANO incorporates these factors through a modality-aware closed-form fusion rule, enabling efficient and effective capability injection.

Table 12: Comparison between DOGE and ANO on InternVL3.5.

| Method | MMMU (%) |
|---|---|
| InternVL3.5-4B (Baseline) | 66.6 |
| DOGE (Wei et al., 2025c) | 52.4 |
| **ANO-4B (Ours)** | **69.6** |

In summary, although ANO and DOGE share similar mathematical roots, their assumptions and effective application domains are fundamentally different: DOGE excels in multi-task LLM merging, while ANO is explicitly designed for heterogeneous multimodal fusion and demonstrates superior empirical performance.

A.22 COMPARISON WITH MODEL-MERGING BASELINES

This section provides a comparison between our method (ANO) and three recent model-merging approaches: OptMerge Wei et al. (2025b), WUDI Cheng et al. (2025), and DOGE Wei et al. (2025c). All methods were evaluated on the MMMU benchmark using the same InternVL3.5-4B backbone.

Table 13: Comparison with model-merging baselines on MMMU.

| Method | MMMU (%) |
|---|---|
| InternVL3.5-4B (Baseline) | 66.6 |
| OptMerge (Wei et al., 2025b) | 51.2 |
| WUDI (Cheng et al., 2025) | 51.8 |
| DOGE (Wei et al., 2025c) | 52.4 |
| **ANO-4B (Ours)** | **69.6** |

Among all evaluated methods, ANO achieves the highest performance. A key observation is that OptMerge focuses on MLLM-to-MLLM merging, while WUDI and DOGE are designed for LLM-to-LLM merging. All three baselines assume same-modality task vectors during fusion. In contrast, our setting requires cross-modal fusion between a vision-conditioned MLLM and a text-only reasoning LLM. This modality mismatch likely accounts for the substantial degradation observed when applying these unimodal-merging baselines directly to heterogeneous multimodal architectures.

By contrast, ANO incorporates modality-aware layer specialization, preserving shallow perceptual pathways while injecting deep reasoning capability, resulting in stable and effective cross-modal capability transfer. These results highlight the advantage of our closed-form, prior-guided layer-wise fusion strategy in multimodal settings.

A.23 CORRELATION-AWARE FUSION AND THE VALIDITY OF THE ORTHOGONALITY ASSUMPTION

This section provides a consolidated discussion of (1) the empirical validity of the near-orthogonality assumption between task vectors, and (2) a correlation-aware extension of our fusion objective.

Our method transfers long-chain-of-thought reasoning from an LLM into an MLLM. In this heterogeneous setting, the two task vectors arise from different modalities, vision-instruction tuning versus pure-text reasoning, and are trained on disjoint data distributions and objectives. Empirically, this leads to near-orthogonality between the task vectors. Across all tested model scales (8B, 14B, 38B), the cosine similarity between the vision task vector and reasoning task vector remains close to zero. This behavior holds consistently across architectures, supporting the validity of the orthogonality assumption in cross-modal fusion.

Although orthogonality holds strongly in practice, we also consider a more general formulation where the task vectors at layer $l$ may have non-zero correlation:

$$c^{(l)} = \left\langle \tau_V^{(l)}, \tau_R^{(l)} \right\rangle \neq 0. \tag{49}$$

In this case, the layer-wise quadratic objective becomes:

$$J_{\text{corr}}^{(l)} = \sum_{t \in \{V,R\}} \frac{\delta_t^{(l)}}{2} \|\tau_t\|^2 \Big[ (1 - \lambda_t)^2 \|\tau_t\|^2 + \sum_{k \neq t} \lambda_k^2 \|\tau_k\|^2 + 2(1 - \lambda_t) \sum_{k \neq t} \lambda_k \, c^{(l)} \Big]. \tag{50}$$

Taking the gradient with respect to $\lambda$ yields a layer-wise linear system:

$$\mathbf{M}^{(l)} \lambda^{(l)} = \mathbf{y}^{(l)}, \tag{51}$$

where both curvature terms $\delta_t^{(l)}$ and correlation $c^{(l)}$ enter explicitly.

While this correlation-aware objective is more general, it introduces several limitations:

- The closed-form solution in Eq. (13) of the main paper no longer holds.
- Curvature terms $\delta_t^{(l)}$ no longer cancel and must be estimated from data, contradicting our training-free objective.
- Computing $c^{(l)}$ for all layers increases computational overhead.

Given that task vectors in our heterogeneous setup are empirically near-orthogonal across all tested architectures and scales, the closed-form ANO fusion provides the best trade-off between accuracy, interpretability, and data-free applicability.

### A.24 ADDITIONAL VALIDATION OF MODALITY PRIOR ACROSS ARCHITECTURES AND MODEL SCALES

This section provides extended evidence for the contribution that shallow decoder layers primarily support visual perception while deeper layers support textual reasoning. To verify the generality of this layer-wise functional specialization, we conducted additional ablations across more architectures and model scales. We evaluated ANO with and without the Modality Prior (MP) on 8B, 15B, and 38B models. As shown in Table 14, incorporating MP consistently yields substantial performance improvements across all tested multimodal benchmarks.

Table 14: Effect of Modality Prior across model scales.

| Model | MMMU | MMMU-Pro | MathVision | WeMath |
|---|---|---|---|---|
| ANO-8B (w/o MP) | 45.2 | 34.0 | 22.9 | 11.1 |
| ANO-8B (w/ MP) | 48.3 | 34.7 | 27.6 | 11.7 |
| ANO-15B (w/o MP) | 58.4 | 41.8 | 28.7 | 31.4 |
| ANO-15B (w/ MP) | 61.3 | 49.4 | 37.2 | 32.3 |
| ANO-38B (w/o MP) | 65.4 | 51.4 | 37.0 | 42.5 |
| ANO-38B (w/ MP) | 69.2 | 56.8 | 39.7 | 47.0 |

We further evaluated two widely used vision-language architectures, LLaVA-OneVision 1.5-4B and Qwen2.5-VL-7B, which were not included in the main paper. As shown in Table 15, the Modality Prior again brings consistent improvements. These results demonstrate that the prior is robust across diverse encoder-decoder designs.

Across all tested model scales (8B-38B) and architectures (InternVL, LLaVA-OneVision, Qwen2.5-VL), the Modality Prior consistently improves multimodal performance. These results reinforce the generality of the "shallow-perception, deep-reasoning" specialization and support its use as a principled prior in our fusion method.

Table 15: Effect of Modality Prior across additional architectures.

| Model | MMMU (%) |
| --- | --- |
| LLaVA-OneVision1.5-4B (w/o MP) | 58.4 |
| LLaVA-OneVision1.5-4B (w/ MP) | 62.9 |
| Qwen2.5-VL-7B (w/o MP) | 54.2 |
| Qwen2.5-VL-7B (w/ MP) | 55.0 |
| InternVL3.5-4B (w/o MP) | 66.6 |
| InternVL3.5-4B (w/ MP) | 69.6 |

