# OpenReview forum: "Training-Free Reasoning and Reflection in MLLMs"
_ICLR.cc/2026/Conference — Submitted to ICLR 2026_

### Official Review · Reviewer_e7Uz · 2025-11-01

**Soundness:** 3
**Presentation:** 3
**Contribution:** 3
**Rating:** 4
**Confidence:** 3

**Summary:**

Built on top of task arithmetic, this paper leverages the finding that shallow decoder layers in MLLMs tend to focus on
visual perception and deeper layers on textual reasoning and proposed layer-wise task vector merging.  The paper derives a closed-form solution to guide layer-wise task vector merging and demonstrated significant model performance improvement on multimodal reasoning tasks.

**Strengths:**

Quality: principled way of using Taylor expansion to derive a closed-form optimal fusion strategy for each layer.


Significance: Proposes an effective and efficient way of combining non-reasoning MLLMs and reasoning LLMs into powerful MLLMs that have strong multimodal reasoning capabilities.

**Weaknesses:**

1.  By heavily weighting the text-based reasoning in deeper layers, the proposed method might have weakened native image-based reasoning pathways. The paper could be strengthened by more experiments and analysis on how the proposed method could affect model's image-based reasoning capability, especially on datasets where the prompt format is image.

2.  "shallow decoder layers focus on visual perception and deeper layers on textual reasoning, is identified and leveraged." could be strengthened by testing on more architectures and across model scales if it were to be included as one of the contributions.

**Questions:**

Could you help clarify what each color in figure 2(a), 4(a) and 5(a) represents?

---

> ### Author Response · Authors · 2025-11-21
>
> We thank reviewer for the constructive comments, We provide our feedbacks as follows:
>
> ### **Q1. About the prompt format is image.**
>
> Thank you for the suggestion.
>
> Regarding the reviewer’s concern about datasets where the entire prompt is presented as an image, we conducted additional experiments on MMMU-Pro-Vision, where the question, problem statement, and answer choices are all image-rendered. We evaluated ANO-4B, and the results show a substantial improvement over the baseline, indicating that our method preserves and even enhances image-based reasoning pathways in this setting.
>
>
> | Method                     | MMMU-Pro-Vision (Acc) |
> |----------------------------|-----------------------:|
> | Idefics3-8B-Llama3         | 15.6 |
> | LLaVA-OneVision-7B         | 18.7 |
> | Qwen2-VL-7B                | 27.0 |
> | InternVL3.5-4B (Baseline)  | 24.1 |
> | **ANO-4B (Ours)**          | **28.6 (+4.5)** |
>
>
> ----
> ### **Q2. About testing on more architectures and across model scales.**
>
> Thank you for the suggestion.
>
> To further validate the contribution that shallow decoder layers focus on visual perception while deeper layers focus on textual reasoning, we conducted additional ablations across more architectures and model scales.
>
> Beyond the models already evaluated in the main paper, we added experiments on two widely used architectures, LLaVA-OneVision 1.5-4B and Qwen2.5-VL-7B. The results consistently show that incorporating the **Modality Prior (MP)** yields substantial performance improvements across all tested architectures and scales.
>
> | Model    | MMMU | MMMU-Pro | MathVision| WeMath|
> |----------------------|------:|---------:|----------:|-----------:|
> | **ANO-8B (w/o MP)**  | 45.2| 34.0| 22.9|11.1 |
> | **ANO-8B (w/ MP)**    | 48.3 | 34.7 |  27.6 | 11.7 |
> | **ANO-15B (w/o MP)** | 58.4 | 41.8 | 28.7 | 31.4 |
> | **ANO-15B (w/ MP)**   | 61.3 |     49.4 |      37.2 |       32.3 |
> | **ANO-38B (w/o MP)** | 65.4 |  51.4 | 37.0 | 42.5 |
> | **ANO-38B (w/ MP)**   | 69.2 |     56.8 |      39.7 |       47.0 |
>
>
> | Model     | MMMU |
> |----------------------|-----------:|
> | **LLaVA-OneVision1.5-4B + Ours (w/o MP)**  | 58.4 |
> | **LLaVA-OneVision1.5-4B + Ours (w/ MP)**    | 62.9 |
> | **Qwen2.5-VL-7B + Ours (w/o MP)**  | 54.2 |
> | **Qwen2.5-VL-7B + Ours (w/ MP)**    | 55.0 |
> | **InternVL3.5-4B (w/o MP)**  | 66.6 |
> | **InternVL3.5-4B (w/ MP)**    | 69.6 |
>
>
> These findings reinforce that the layer-wise functional specialization is not specific to a single model family, but is a stable property that can be effectively leveraged by our fusion method.
>
> **We have added a dedicated subsection in Appendix A24.**

---

> ### Author Response · Authors · 2025-11-28
>
> ### **Q3. About each color in figures 2(a), 4(a,) and 5(a) represents.**
>
> We apologize for the earlier omission.
>
> In Figures 2(a), 4(a), and 5(a), each color corresponds to the visual attention of a different output token. For a given prompt, the model generates multiple output tokens, and we use distinct colors to visualize the token-wise visual attention patterns produced at each decoding step.
>
> If you have any further questions, we would be happy to discuss them.

---

### Official Review · Reviewer_Aoes · 2025-11-02

**Soundness:** 2
**Presentation:** 3
**Contribution:** 2
**Rating:** 4
**Confidence:** 3

**Summary:**

This paper introduces ANO, a training-free method for integrating reasoning and reflection capabilities into off-the-shelf Multimodal Large Language Models (MLLMs) by merging them with reasoning-specialized LLMs.  The approach is based on two key insights: (1) the functional specialization of shallow and deep decoder layers in MLLMs (visual perception vs. textual reasoning), and (2) the near-orthogonality of task vectors from vision and reasoning fine-tuning.  The authors propose a layer-wise, Taylor-derived closed-form fusion mechanism that optimally combines task vectors without additional training.

**Strengths:**

- The derivation of closed-form fusion weights using Taylor expansion and NTK linearization is rigorous and well-motivated.
- Comprehensive experiments across multiple benchmarks and model scales (8B to 38B) show consistent and significant improvements over strong baselines.
- The paper includes thorough ablations to validate the contribution of each component (e.g., modality priors, layer-wise fusion).

**Weaknesses:**

- The method relies heavily on the orthogonality of task vectors and the NTK linearization assumption, which may not hold universally, especially for smaller models or non-standard architectures. The authors are encouraged to provide results on more mainstream architectures such as LLaVA, Qwen, etc. Have the authors considered the potential negative impact of task vector interference when the orthogonality assumption is violated?  Are there fallback mechanisms?
- The approach is only validated on vision-language tasks.  Its applicability to other modalities (e.g., video, structured data) remains unverified.

**Questions:**

Please refer to the weaknesses part.

---

> ### Author Response · Authors · 2025-11-21
>
> We thank reviewer for the constructive comments, We provide our feedbacks as follows:
>
> ### **Q1. About more mainstream architectures.**
>
> Thank you for the suggestion.
>
> Regarding mainstream architectures, we already report results on three families (Idefics, NVILA, InternVL) across multiple scales.
>
> To further address the reviewer’s concern, we additionally evaluated **InternVL3.5-4B**, **LLaVA-OneVision1.5-4B**, and **Qwen2.5-VL-7B** on the MMMU benchmark. The results show that **our method consistently improves performance** across both architectures and scales, confirming that ANO does not rely on a specific backbone.
>
> | Architecture / Scale            | w/o ANO | w/ ANO (Ours) |
> |---------------------------------|-------------:|-------:|
> | InternVL3.5-4B |66.6  | **69.6** |
> | LLaVA-OneVision-1.5-4B        |   52.7  |      **62.9** |
> | Qwen2.5-VL-7B                 |   53.9  |      **55.0** |
>
> With these additions, ANO has now been evaluated across 6 architectures and 5 model scales, showing consistently positive gains.
>
> ---
> ### **Q2. About the orthogonality of task vectors and the NTK linearization assumption.**
>
> Thank you for the suggestion.
>
> Our work focuses on transferring LLM-style long-chain reasoning and reflection into MLLMs. In this setting, the vision-task vector and reasoning-task vector are obtained from different modalities and training distributions. Empirically, this leads to **near-orthogonality almost always holding in practice**, as the two vectors capture fundamentally different directions in parameter space. We provide orthogonality visualizations for 8B/15B/38B models in the main paper.
>
> In addition, in the revised version, we include a quantitative analysis of the **NTK linearization assumption** in Appendix A20, showing that multimodal decoders operate very close to the linearized regime. These results jointly validate that both assumptions (near-orthogonality and NTK-style linearity) hold strongly in our multimodal fusion setting.
>
> **`About fallback mechanisms.`**
>
> For completeness, we provide a correlation-aware extension.
>
> When the inner product $(c^{(l)}=\langle \tau_V^{(l)},\tau_R^{(l)}\rangle\neq 0)$, the layer-wise objective leads to a linear system $(\mathbf{M}\lambda=\mathbf{y})$ that accounts for curvature and cross-task correlation. While more general, this formulation:
> - removes the clean closed-form update of ANO,
> - requires data-dependent curvature estimation (violating the training-free goal), and
> - introduces additional computational overhead.
>
> Given that correlation is empirically negligible across all tested architectures and scales, the closed-form ANO formulation remains the most accurate, efficient, and practical.
>
> **We have added a dedicated subsection in Appendix A23.**
>
> ---
> ### **Q3. About applicability to other modalities.**
>
> Thank you for the suggestion.
>
> To assess applicability beyond vision–language tasks, we additionally evaluated ANO-4B on the VideoMMMU benchmark. As shown in the Table below, our method yields clear and consistent improvements in the video modality as well, demonstrating that the proposed fusion strategy generalizes beyond static images.
>
> | Model                         | VideoMMMU (%) |
> |-------------------------------|---------------:|
> | InternVL3.5-4B (Baseline)     | 42.9          |
> | **ANO-4B (Ours, training-free)** | **45.4**      |
>
> Regarding structured data, our primary goal is to transfer LLM reasoning into multimodal models, and we therefore focus on modalities where visual perception interacts with high-level reasoning. We are not aware of widely adopted multimodal benchmarks involving structured data. If the reviewer has specific tasks or datasets in mind, we would be happy to discuss them further.

---

### Official Review · Reviewer_pwa5 · 2025-11-03

**Soundness:** 2
**Presentation:** 2
**Contribution:** 2
**Rating:** 4
**Confidence:** 3

**Summary:**

This paper proposes ANO, a training-agnostic, R1-style multimodal model fusion method. ANO leverages task vectors and second-order Taylor/NTK linearization to derive closed-form fusion weights at every decoder layer. It fits a modality prior from the exponentially decaying visual attention across layers to preserve visual perception in shallow layers while injecting language reasoning and self-reflection capabilities in deeper layers, yielding a strong reasoning MLLM without additional data or training.

**Strengths:**

The method requires no gradient updates, reinforcement learning, or extra labeled data. Fusion coefficients are given in closed form, avoiding grid search and validation-set tuning. Under NTK linearization and approximate orthogonality of task vectors, a second-order Taylor expansion yields closed-form fusion weights whose dependence only on layer-wise task-vector norms is both concise and interpretable. Empirical results demonstrate successful transfer of R1-like reasoning and self-reflection behaviors.

**Weaknesses:**

1. A central insight that shallow layers handle perception while deep layers handle reasoning (Fig. 2) has already been articulated in [1], which diminishes the contribution.
2. The paper lacks discussion and comparison with closely related work. Both the proposed method and [2] reformulate differences between the merged model and task-specific models via Taylor approximation (paired with NTK linearization and high-dimensional approximate orthogonality) into a data-free computable objective. In essence, replacing the true loss in a local neighborhood with a low-order approximation. The assumptions (NTK regime, small perturbations, near-orthogonality) are shared. Although this paper uses a second-order expansion, the similarities are substantial; why is [2] not discussed?
3. Missing comparisons against recent model-merging baselines. What happens if the proposed merging method is replaced with alternatives such as DOGE [2], WUDI [3], or OptMerge [4]?
4. Performance on state-of-the-art MLLMs (e.g., Qwen2.5-VL and Qwen3-VL) is not reported.

[1] Bring Reason to Vision: Understanding Perception and Reasoning through Model Merging. ICML 2025.

[2] Modeling Multi-Task Model Merging as Adaptive Projective Gradient Descent. ICML 2025.

[3] Whoever Started the Interference Should End It: Guiding Data-Free Model Merging via Task Vectors. ICML 2025.

[4] Unifying Multimodal Large Language Model Capabilities and Modalities via Model Merging. arXiv2505.

**Questions:**

N/A

---

> ### Author Response · Authors · 2025-11-21
>
> We thank reviewer for the constructive comments, We provide our feedbacks as follows:
>
> ### **Q1. About shallow layers handle perception while deep layers handle reasoning.**
>
> Thank you for the suggestion.
>
> While [1] indeed reports a descriptive observation that shallow layers tend to emphasize perception and deeper layers emphasize reasoning, their analysis remains purely observational. **The phenomenon is stated but not turned into any algorithmic mechanism or used to improve multimodal model performance**.
>
> In contrast, our contribution is not the observation itself, but a principled method that uses this layer-wise functional differentiation to enable effective multimodal fusion. We (1) conduct a more fine-grained empirical analysis confirming the pattern in multimodal decoders, and (2) introduce a layer-wise closed-form fusion strategy that injects reasoning abilities into MLLMs while maximally preserving shallow-layer visual competence. This hierarchical fusion mechanism is new and central to the performance gains demonstrated in the paper.
>
> Thus, while the high-level intuition is known, our method is the first to turn this intuition into a practical, theoretically grounded fusion algorithm for training-free multimodal reasoning.
>
> In fact, we have already provided a comprehensive comparison with [1] in the main paper (Table 2). As shown in Table 2 (reproduced below for convenience), we directly compare ANO with [1]. ANO significantly outperforms both the NVILA baseline and [1]:
>
> | Method (NVILA-15B)              |     MMMU | MMMU-Pro | MathVision |   WeMath |      Avg |
> | ------------------------------- | -------: | -------: | ---------: | -------: | -------: |
> | NVILA* (Baseline)               |     53.2 |     36.2 |       23.2 |     31.1 |     35.9 |
> | VLM-Merging [1] |     53.6 |     36.5 |       24.5 |     31.1 |     36.4 |
> | **ANO-15B (Ours)**              | **61.3** | **49.4** |   **37.2** | **32.3** | **45.1** |
>
> ---
> ### **Q2. About related work [2].**
>
> Thank you for the suggestion.
>
> We agree that both our method and [2] build on a local Taylor approximation with NTK-style linearization. The omission was purely due to timing; the acceptance (ICML2025) of [2] was announced around the same period our submission was finalized.
>
> **While the theoretical starting point is similar, the problem settings, strengths, and limitations of the two approaches differ substantially. [2] is designed for multi-task merging, and its strength lies in an optimization framework that can reconcile multiple conflicting task vectors. Our method targets a different goal, cross-modal capability injection, and leverages functional specialization in MLLMs.**
>
> To directly assess whether methods such as DOGE [2] can transfer to the multimodal setting, we applied DOGE to the same architecture used for ANO-4B (InternVL3.5). DOGE dramatically degrades performance on MMMU:
>
> | Method  | MMMU (%) |
> | - | - |
> |  InternVL3.5-4B (Baseline) |     66.6 |
> | DOGE [2] |     52.4 |
> |  **ANO-4B (Ours)** | **69.6** |
>
> The degradation indicates that DOGE’s assumptions, designed for merging task vectors in a unimodal LLM context, do not hold in heterogeneous multimodal architectures, where cross-modal interactions and layer specialization must be explicitly modeled.
>
> **We have added a dedicated subsection comparing the two methods and clarifying their respective strengths and limitations in Appendix A21.**

---

> > ### Author Response · Authors · 2025-11-21
> >
> > ### **Q3. About recent model-merging baselines.**
> >
> > Thank you for the suggestion.
> >
> > Following the reviewer’s advice, we added comparisons with three recent model-merging baselines, DOGE [2], WUDI [3], and OptMerge [4], on the MMMU benchmark.
> >
> > |    Method | MMMU (%) |
> > | ----------------------: | -------: |
> > | InternVL3.5-4B (Baseline) |     66.6 |
> > | OptMerge | 51.2|
> > | WUDI | 51.8|
> > | DOGE |     52.4 |
> > |  **ANO-4B (Ours)** | **69.6** |
> >
> > Across all settings, our method achieves the best performance. We note that OptMerge focuses on MLLM-to-MLLM fusion, while WUDI and DOGE are designed for LLM-to-LLM merging; all three assume the same-modality task vectors. In contrast, our setting requires cross-modal fusion between a vision-conditioned MLLM and a text-only reasoning LLM. This modality mismatch likely explains why these unimodal merging baselines degrade significantly when applied to heterogeneous multimodal architectures.
> >
> > Our closed-form, prior-guided layer-wise fusion explicitly models modality specialization (perception vs. reasoning), enabling stable and effective cross-modal capability injection. **These results have been included in the revised Appendix A22.**
> >
> > ---
> > ### **Q4. About the performance of Qwen2.5-VL and Qwen3-VL.**
> >
> > Thank you for the suggestion.
> >
> > In fact, we have reported results on a strong state-of-the-art MLLM: Table 11 in Appendix A.10 provides a detailed comparison with Qwen2.5-VL, showing that our method yields substantial improvements over this baseline without any additional training.
> >
> > Regarding Qwen3-VL, it was released after our submission deadline and, therefore, could not be included in the initial manuscript. To address the reviewer’s concern, we additionally evaluated our method on Qwen3-VL-4B. The results show that our ANO-4B model achieves performance comparable to Qwen3-VL 4B, demonstrating that the proposed fusion strategy remains effective even against the latest MLLMs.
> >
> > | Model                     | MMMU | MathVista | MathVision | WeMath |
> > |---------------------------|-----:|----------:|------------:|-------:|
> > | **Qwen2.5-VL-7B**         | 53.9 | 67.8      | 25.4        | 35.2   |
> > | **Qwen2.5-VL-72B**        | 68.2 | 74.2      | 39.3        | 49.1   |
> > | **Qwen3-VL-4B (Reproduction)**           | 68.8 | 77.5      | 58.1        |   –    |
> > | **InternVL3.5-4B (Baseline)** | 66.6 | 77.1      | 54.4        | 50.1   |
> > | **ANO-4B (Ours, training-free)** | 69.6 | 77.6 | 55.1 | 51.9 |

---

### Official Review · Reviewer_e8sc · 2025-11-07

**Soundness:** 2
**Presentation:** 3
**Contribution:** 3
**Rating:** 6
**Confidence:** 4

**Summary:**

The paper proposes a training-free, layer-wise merge of two models: a vision-grounded MLLM and a text-reasoning LLM. Instead of fine-tuning, they combine weights per decoder layer using a closed-form rule derived from a local second-order (Taylor) approximation; shallow layers are biased to keep perception, deeper layers to inject reasoning via an attention-based prior fit from the model’s own layer-wise visual attention. The idea is simple to deploy (no gradients, no grid search) and shows notable gains on multimodal reasoning (e.g., 69.2 on MMMU with the 38B variant, beating strong open baselines and matching/exceeding proprietary systems on that benchmark).

My reservations are about assumptions and scope. The derivation leans on NTK-style linearization and near-orthogonality of task vectors; both can be only approximately true in finite multimodal transformers, and the paper itself shows performance drops when task vectors are less orthogonal. The modality prior is fit on generic images (e.g., MSCOCO) and most wins are on math/diagram-heavy benchmarks, so it’s unclear how robust the scheme is across architectures or domains that don’t share the same attention profile. In short: novel and practical, with strong initial results, but it should have stronger sensitivity checks (when assumptions weaken) and broader evaluations before concluding it generalizes widely.

**Strengths:**

- The paper replaces heuristic task-arithmetic with a per-layer merger derived from a second-order Taylor approximation under NTK-style linearization, yielding analytic coefficients (their Eq. (13), prior-weighted in Eq. (15)) that require no grid search or supervision.

- The attention-guided prior is fit to the model’s observed decay of visual attention across depth and then used to bias fusion via a simple exponential schedule—tightening the link between empirical signal and architectural choice.

- The authors compute per-layer cosine similarity between vision and reasoning deltas and show near-orthogonality (justifying the decoupled quadratic form), and they further stress-test cases where this assumption weakens, noting degraded performance—usefully delineating the method’s operating regime.

- Using MME, the full method (with the prior) stays close to a perceptual “upper bound” while improving reasoning, and contrasts cleanly with the ablation without the prior—making the impact on perception explicit and measurable.

**Weaknesses:**

- The closed-form per-layer coefficients rest on NTK-style linearization and an isotropic Hessian surrogate, but the paper provides no error bounds or diagnostics quantifying deviation from these ideals in finite-width, multimodal transformers; optimality can drift when curvature/anisotropy is non-negligible.

- The derivation effectively requires near-orthogonality between the vision and reasoning deltas; when vectors correlate, performance materially degrades, and the method lacks a correlation-aware fusion or mitigation strategy (e.g., whitening/rotation, damping).

- The exponential attention-decay prior is fitted on ~1k MSCOCO images and then applied to math/diagram reasoning; robustness under domain or encoder shifts is not established, making the prior potentially domain-sensitive outside the measured regime.

- Most wins are on MMMU/MMMU-Pro/MathVision/WeMath (text-in-image/diagram heavy). Absent broader benchmarks (document OCR beyond math, chart QA outside math, natural-image long-context VQA, shifted domains), generality remains unproven.

- Results are reported at single settings; there are no sensitivity surfaces versus global fusion strength or prior slope, nor curves versus induced task-vector correlation. Without these, practitioners lack clear guardrails for accuracy–perception trade-offs and failure modes.

**Questions:**

- Can you quantify how closely the multimodal decoder operates in the NTK-style linear regime assumed for your second-order Taylor derivation

- Your derivation and Eq. (13)/(15) rely on near-orthogonality of task vectors, yet Appendix stress tests show degradation when cosine similarity rises. Can you provide a correlation-aware fusion

- how stable are the fitted decay parameters across domains (diagrams, charts, UI screenshots) and encoders? Please report per-domain priors and sensitivity of accuracy to the prior slope α.

- How would you fuse non-homologous backbones or differing tokenizer/vision-encoder stacks (e.g., alignment layers/adapters)? Any empirical evidence beyond the specific pairings in Appendix A.4?

---

> ### Author Response · Authors · 2025-11-21
>
> We thank reviewer for the constructive comments. We provide our feedbacks as follows:
>
> ### **Q1. About the NTK-style linear regime assumed for the second-order Taylor derivation.**
>
> Thank you for the suggestion.
>
> To quantify the NTK-style linearization in practice, we ran a direct diagnostic on three representative multimodal decoders (Idefics3-Llama 8B, NVILA 15B, InternVL2.5 38B). For each model we linearly interpolated parameters $\theta(\alpha) = (1-\alpha)\theta_0 + \alpha\theta_t$, evaluated the flattened logits $f(\theta(\alpha))$ on several prompts, and computed the projection
>
> \begin{equation}
> s(\alpha) = \frac{(f(\theta(\alpha)) - f(\theta_0)) \cdot \Delta}{\|\Delta\|^2}, \qquad \Delta = f(\theta_t) - f(\theta_0).
> \end{equation}
>
> together with the normalized residual $R(\alpha)/\|\Delta\|$. Under exact NTK linearization, $s(\alpha)\approx\alpha$.
>
> **We have added the visualization results and analysis to Appendix A20.** The provided figures show that across prompts and models, $s(\alpha)$ closely follows the reference line $y=\alpha$, and the normalized residuals remain small and decrease with model scale. These visualizations directly quantify the deviation from the NTK linear regime: $\lvert s(\alpha)-\alpha\rvert$ and $R(\alpha)/\|\Delta\|$ serve as explicit, data-driven measures of linearization error.
>
> --------
> ### **Q2. About correlation-aware fusion**
>
> Thank you for the suggestion.
>
> It needs to be emphasized that our work focuses on transferring long-chain-of-thought reasoning and reflection abilities from LLMs to MLLMs. Since the two task vectors are learned from fundamentally different modalities, vision-instruction tuning versus pure-text reasoning, they are trained on disjoint data distributions and objectives. **In our experience, this heterogeneous setup almost always yields near-orthogonal task vectors. This is consistent with our empirical observations: in the main paper and Appendix, cosine similarity for 8B, 15B, and 38B models remains close to zero**. Across model scales and architectures, the near-orthogonal pattern consistently holds, supporting the validity of this assumption in cross-modal fusion.
>
> Correlation-aware fusion.
>
> We appreciate the reviewer’s suggestion to consider scenarios where task vectors may not be perfectly orthogonal. To explore this direction, we provide a correlation-aware formulation as a potential extension for future work. When the inner product
>
> $\(c^{(l)} = \langle \tau_V^{(l)}, \tau_R^{(l)} \rangle \neq 0\)$,
> the layer-wise objective becomes:
>
> \begin{equation}
> J_{\mathrm{corr}}^{(l)}
> = \sum_{t\in\{V,R\}}
> \frac{\delta_t^{(l)}}{2}\\big\|\tau_t^{(l)}\big\|^2 \Bigg( (1-\lambda_t)^2\big\|\tau_t^{(l)}\big\|^2 + \sum_{k\neq t}\lambda_k^2\big\|\tau_k^{(l)}\big\|^2 + 2(1-\lambda_t)\,c^{(l)}\sum_{k\neq t}\lambda_k\Bigg)
> \end{equation}
>
> Setting $\(\nabla J_{\mathrm{corr}}^{(l)} = 0\)$ yields a linear system $\(\mathbf{M}\lambda = \mathbf{y}\)$, where curvature $\(\delta_t^{(l)}\)$ and correlation $\(c^{(l)}\)$ appear explicitly.
>
> While this correlation-aware formulation is more general, it brings several practical drawbacks:
> - it removes the clean closed-form solution in Eq. (13);
> - curvature terms no longer cancel and would require data-dependent estimation, conflicting with our training-free objective;
> - computing $\(c^{(l)}\)$ for all layers increases overhead.
>
> Given that task vectors in our heterogeneous (Vision + Reasoning) setting are empirically near-orthogonal across all tested scales, ANO provides the best balance of accuracy, interpretability, and data-free applicability.
>
> **We have added a dedicated subsection in Appendix A23.**

---

> > ### Author Response · Authors · 2025-11-21
> >
> > ### **Q3. About per-domain priors and the sensitivity of accuracy to the prior slope $\alpha$.**
> >
> > Thank you for the suggestion.
> >
> > We originally fit the exponential decay prior on **MSCOCO** to avoid domain leakage (MSCOCO is broad and non-specialized). To verify stability, we re-fitted ($\alpha$) separately on two domain subsets (Charts: ChartQA, UI screenshots: ScreenSpot) and evaluated each prior on the multi-domain MMMU benchmark. We were unable to find a dataset exclusively targeting Diagrams, and therefore report the two domains that have clean, dedicated subsets.
> >
> > | Domain (for fitting prior) | Dataset  | Fitted decay α |
> > | -------------------------- | -------- | -------------- |
> > | General (baseline)         | MSCOCO   | **0.108**      |
> > | Charts                   | ChartQA  | **0.115**      |
> > | UI Screenshots             | ScreenSpot | **0.111**      |
> >
> >
> > | Prior source           | MMMU Accuracy (%) |  vs. MSCOCO prior |
> > | ---------------------- | ----------------- | ------------------ |
> > | MSCOCO prior (default) | **42.3**          | –                  |
> > | ChartQA-derived prior  | **42.5**          |       +0.2         |
> > | ScreenQA-derived prior | **42.2**          |     –0.1           |
> >
> > Across both specialized domains (charts, UI), the fitted decay slopes remain within a narrow band. These variations fall inside the expected statistical noise range, indicating that our method is robust to domain shifts and that the exponential decay prior is highly stable across domains.
> >
> > We additionally applied the MSCOCO priors to ANO-4B (InternVL) and evaluated on two benchmarks, ChartQA and OCRBench. Both tasks involve domain-specific visual structures quite different from natural-image datasets like MSCOCO, and our method still yields consistent performance improvements under these domain shifts.
> >
> > | Method                    |  ChartQA | OCRBench |
> > | ------------------------- | -------: | -------: |
> > | InternVL3.5-4B (Baseline) |     86.0 |      822 |
> > | **ANO-4B (Ours)**         | **86.5** |  **827** |
> >
> > -----
> > ### **Q4. About fuse non-homologous backbones.**
> >
> > Thank you for the suggestion.
> >
> > Model fusion fundamentally relies on homologous architectures (identical layer structure and parameter shapes), which is a standard assumption in task-vector and parameter-arithmetic literature. Beyond the three model pairs reported in Appendix A.4, we additionally tested three further architectures with different sizes on MMMU benchmark, ANO-4B (InternVL3.5), ANO-4B (LLaVA-OneVision-1.5) and ANO-7B (QwenVL-2.5), and both show consistent improvements under our fusion procedure.
> >
> > | Architecture | w/o ANO | w/ ANO (Ours) |
> > |---------------------------------|-------------:|-------:|
> > | InternVL3.5-4B |66.6  | **69.6** |
> > | LLaVA-OneVision-1.5-4B        |   52.7  |      **62.9** |
> > | Qwen2.5-VL-7B                 |   53.9  |      **55.0** |
> >
> > **With these additions, ANO has now been evaluated across 6 architectures and 5 model scales, showing consistently positive gains.**
> >
> > Importantly, when we deliberately attempted to fuse non-homologous backbones, such as Qwen vs. LLaMA, or models using mismatched tokenizer or vision-encoder stacks, the fused model produced no valid outputs. This failure mode confirms that architectural homology is a hard requirement for parameter-space fusion. Extending ANO to heterogeneous backbones would require explicit alignment layers (e.g., learned linear adapters or projection modules), which we leave as future work.

---

### Meta-Review · Area_Chair_SMFh · 2026-01-04

**Summary:**

The paper proposes a model merging method to combine a reasoning LLM and an MLLM so that the merged model is good at multimodal reasoning. The concerns from reviewers are impractical assumptions, insufficient experiment and potentially weak generalization. The authors have posted the rebuttal to address those concerns but I believe some of them are still unaddressed. Specifically, when the proposed ANO is evaluated on broader benchmarks, the baselines should be model merging methods instead of a single model as ANO uses two strong models. Otherwise, it makes the comparison not fair. Thus I don't think the generalization concern is well addressed. Additionally, the modality prior design needs more analysis as this component is the main reason why ANO works well, as indicated by Tab. 2. Overall, it is a good paper with interesting empirical results, but given the existing weaknesses in evaluation and analysis, I would recommend rejection.

**Reviewer Concerns:**

The following concerns are addressed:


1.	The assumptions might not hold in practice (e8sc, Aoes)
2.	Robustness under distribution shift is a concern (e8sc)
3.	The novelty is limited (pwa5)
4.	Missing comparisons with related work and baselines (pwa5)
5.	Performance on state-of-the-art MLLMs is not reported (pwa5)
6.	Lack of validation on other modalities (Aoes)
7.	Lack of analysis on the image-based reasoning capability (e7Uz)
8.	More experiments are needed to verify the argument on shallow and deep layer reasoning (e7Uz)

The following concerns are outstanding:


1.	Generalization on broader benchmarks might be weak (e8sc)
2.	Lack of analysis on hyperparameter analysis and failure modes (e8sc)

**Reviewer Scores:**

None of them has responded or indicated to increase the score, I would assume the scores will keep the same.

---

### Decision · Program_Chairs · 2026-01-26

Reject